# Relation Augmented Preferential Bayesian Optimization via Preference Propagation

## Abstract

In black-box optimization, when directly evaluating the function values of solutions is very costly or infeasible, access to the objective function is often limited to comparing pairs of solutions, which yields dueling black-box optimization. Dueling optimization is solely based on pairwise preferences, and thus notably reduces cost compared with function value based methods such as Bayesian optimization. However, an optimization performance gap obviously exists between dueling based and function value based methods. This is mainly due to that most existing dueling optimization methods do not make full use of the pairwise preferences collected. To fill this gap, this paper proposes relation augmented preferential Bayesian optimization (RAPBO) via preference propagation. By considering solution similarity, RAPBO aims to uncover the potential preferential relations between solutions within different preferences through the proposed preferential relation propagation technique. Specifically, RAPBO first clusters solutions using a Gaussian mixture model. After obtaining the solution set with the highest intra-cluster similarity, RAPBO utilizes a directed hypergraph to model the potential relations between solutions, thereby realizing relation augmentation. Extensive experiments are conducted on both synthetic functions and real-world tasks such as motion control and spacecraft trajectory optimization. The experimental results disclose the satisfactory accuracy of augmented preferences in RAPBO, and show the superiority of RAPBO compared with existing dueling optimization methods. Notably, it is verified that, under the same evaluation cost budget, RAPBO is competitive with or even surpass the function value based Bayesian optimization methods with respect to optimization performance. The codes can be found in `https://anonymous.4open.science/r/RAPBO-E15F`.

## 1 Introduction

Black-box optimization (Conn et al., 2009; Liu et al., 2022), also termed as derivative-free optimization, is a class of optimization methods designed for situations where the objective function is unknown, complex, or expensive to evaluate. It enables global search for the optimal solution, with Bayesian optimization (BO) (Garnett, 2023; Mei et al., 2023; Shahriari et al., 2016) as a representative. Due to the significant advantages and progress of black-box optimization, it has been widely applied in fields such as chemical synthesis (Shields et al., 2021), machine learning (Freund & Schapire, 1997; Elsken et al., 2019) and reinforcement learning (Qian & Yu, 2021).

In traditional black-box optimization, evaluating the numerical objective function values is typically necessary. However, in many real-world scenarios, acquiring the objective function values can be extremely costly or entirely infeasible (Brochu et al., 2010). It has been found that comparing two solutions by preferences is relatively cheaper than scoring solutions (Kahneman & Tversky, 1979), such as in A/B tests (Siroker & Koomen, 2013). Thus, *dueling or preferential optimization* has been developed as an easier and cheaper alternative, e.g., preferential Bayesian optimization (PBO) (González et al., 2017). Instead of relying on function values, dueling optimization leverages pairwise preferences (i.e., which solution is preferred) to guide the optimization process, making it easier and cheaper in scenarios where evaluating objective function values is costly or infeasible. Dueling optimization has been successfully applied in a wide range of fields, such as visual design optimization (Koyama et al., 2020) and robotic gait optimization (Li et al., 2021), showcasing its adaptability and effectiveness across various domains.

However, due to that most existing dueling optimization methods have typically made simple use of pairwise preferences, a obvious optimization performance gap exists between dueling or preference based methods such as PBO and function value based methods such as BO. It is obviously that insufficient utilization of pairwise preferences may significantly impact the performance of dueling optimization, while making a fuller use of the preferences can improve the optimization process.

**Problem.** Although dueling optimization can optimize using only low-cost pairwise preferences, the insufficient exploitation of preferences of existing methods could significantly limit the optimization performance of dueling optimization, e.g., preferential Bayesian optimization. This leads to an obvious optimization performance gap between preference based and function value based methods.

**Contribution.** This paper aims to fill the optimization performance gap between preference based and function value based methods, and answer whether the preference based methods can match or even surpass the performance of function value based methods. To this end, we propose the relation augmented preferential Bayesian optimization (RAPBO) method via preference propagation. RAPBO aims to uncover the potential preferential relations between different preferences through the proposed preferential relation propagation technique based on solution similarity. The experimental results reveal that the preferences augmented by the preference propagation technique achieve satisfactory accuracy and verify its superiority over existing dueling optimization methods. Notably, it is verified that, within the same evaluation cost budget, the performance of RAPBO can match and even surpass that of function value based Bayesian optimization methods.

The following sections provide an overview of related work and essential preliminaries, detail the proposed RAPBO method, present the experimental results, and conclude the paper.

## 2 RELATED WORK

This section provides a brief overview of the related work, including preferential Bayesian optimization and hypergraph, to explain the necessary preliminary knowledge and notation.

### 2.1 PREFERENTIAL BAYESIAN OPTIMIZATION

To extend BO to scenarios where direct access to the objective function is unavailable, but information about user preferences can be obtained, González et al. (2017) propose a framework called preferential Bayesian optimization (PBO). PBO leverages pairwise preferences to fit a Gaussian process (GP) (Rasmussen & Williams, 2006) within preference function domain. The PBO employs the dueling-Thompson Sampling (DTS) to determine the potential optimal solution and the solution with high uncertainty as candidates for the next duel. Benavoli et al. (2021) prove that the true posterior distribution of the preference function is a skewed Gaussian process (SkewGP), and incorporate SkewGP to enhance the performance of PBO. Based on the work of Benavoli et al. (2021), Takeno et al. (2023) propose a practical method, HB, which ensures high computational efficiency and low sample complexity. Due to the lack of theoretical guarantees for most acquisition functions in PBO, Astudillo et al. (2023) introduce qEUBO, a promising acquisition function with a grounded decision-theoretic justification. Guided by the optimism principle, POP-BO (Xu et al., 2024) constructs a confidence set from preferences and employs an optimistic strategy that ensures a bound on cumulative regret, enabling it to effectively report an estimated best solution with guaranteed convergence. To address the dimensionality issue exacerbated by modeling the preference function, PE-DBO (Zhang et al., 2023) extends the concept of intrinsic effective dimensionality to preference function. Despite these advancements, these methods still do not fully utilize the available pairwise preferences, which continues to impact the performance of dueling optimization.

Instead of constructing a surrogate model to fit the preference function, Sui et al. (2017) and Xu et al. (2020) respectively propose kernel-self-sparring (KSS) and comp-GP-UCB (COMP-UCB). KSS uses a GP to model the function, where the value represents the probability of one solution beating the optimal solution, rather than modeling a preference function. COMP-UCB employs the Borda function, inspired by the Borda score (Sui et al., 2018), to replace the preference function and regards the average performance of all solutions as the basis for comparison. While these methods simplify the dueling optimization problems compared to the methods that model the preference function, they may still face challenges caused by the insufficient utilization of pairwise preferences, leading to performance that cannot match that of function value based methods.

## 2.2 HYPERGRAPH REPRESENTATION

Hypergraphs (Bretto, 2013) are mathematical models that extend the classical graph structure. In a traditional graph, edges are binary relations connecting two vertices, while a hypergraph allows edges to connect multiple vertices, and these edges are called hyperedges. This characteristic enables hypergraphs to naturally represent more complex, higher-order relationships and interactions, particularly excelling in modeling multi-party interactions. Various algorithms, such as hypergraph partitioning (Papa & Markov, 2007) and hypergraph clustering (Zhou et al., 2006), have been developed to efficiently process hypergraph structures, further enhancing their applicability in large-scale data-driven tasks. Consequently, hypergraphs are widely used in fields such as machine learning (Gao et al., 2022), data mining (Ji et al., 2020), and social network analysis (Lin et al., 2009).

# 3 PRELIMINARIES

## 3.1 DUELING OPTIMIZATION

Consider a black-box function $f : \mathcal{X} \to \mathbb{R}$, where $\mathcal{X} \subset \mathbb{R}^D$, which is costly to evaluate. The goal of global optimization is to find the optimal solution $\boldsymbol{x}^* = \operatorname{argmax}_{\boldsymbol{x} \in \mathcal{X}} f(\boldsymbol{x})$ in a $D$-dimensional continuous *solution space*. Instead of directly evaluating numerical function values, the objective function is evaluated by comparing pairs of solutions $(\boldsymbol{x}, \boldsymbol{x}')$, i.e., duels. An human oracle provides feedback on which solution in a duel is better, yielding binary information (i.e., 0 for $\boldsymbol{x}'$ and 1 for $\boldsymbol{x}$). This type of feedback is referred to as preference, and only these preferences will be used during the optimization process. Throughout this paper, each duel is treated as a coloum vector, represented by $[\boldsymbol{x}; \boldsymbol{x}'] \in \mathbb{R}^{2D}$, where the space with dimension $2D$ is called *dueling solution space*.

**Preference Function.** In dueling optimization, the feedback from a comparison between two solutions $[\boldsymbol{x}; \boldsymbol{x}']$ is treated as a stochastic process. This feedback is sampled from a Bernoulli distribution, where the probability reflects the likelihood that solution $\boldsymbol{x}$ is preferred over $\boldsymbol{x}'$. Under the assumption that the probability of solution $\boldsymbol{x}$ being preferred over $\boldsymbol{x}'$ is positively correlated with the difference in their objective function values, i.e., $P(\boldsymbol{x} \succ \boldsymbol{x}') \propto f(\boldsymbol{x}) - f(\boldsymbol{x}')$, and the logistic function is commonly used to convert this difference into a probability. Therefore, the preference function in the dueling solution space can be formulated as

$$\pi_f([\boldsymbol{x}; \boldsymbol{x}']) = P(\boldsymbol{x} \succ \boldsymbol{x}') = \frac{1}{1 + e^{-[f(\boldsymbol{x}) - f(\boldsymbol{x}')]}}, \tag{1}$$

where $\pi_f([\boldsymbol{x}; \boldsymbol{x}'])$ represents the probability that solution $\boldsymbol{x}$ is preferred over solution $\boldsymbol{x}'$ in the dueling solution space.

**Copeland Score.** To find the optimal solution $\boldsymbol{x}^*$, we introduce the concept of the *Condorcet winner*, an extension from multi-armed bandit tasks, which is the solution that outperforms all others. However, in dueling optimization, a strict Condorcet winner cannot be obtained, so the solution with the highest Copeland score (González et al., 2017) is selected as the best one. Due to the objective function is continuous, the normalized Copeland score is defined as

$$S(\boldsymbol{x}) = \text{Vol}(\mathcal{X})^{-1} \int_{\mathcal{X}} \mathbb{I}_{\{\pi_f([\boldsymbol{x}; \boldsymbol{x}']) \geq 0.5\}} \, d\boldsymbol{x}', \tag{2}$$

where $\text{Vol}(\mathcal{X})^{-1} = \int_{\mathcal{X}} 1 d\boldsymbol{x}'$ is a normalizing constant that ensures $S(\boldsymbol{x})$ is in the $[0, 1]$ range and $\mathbb{I}_{\{\cdot\}}$ is the indicator function. For the optimal solution $\boldsymbol{x}^*$, $\pi_f([\boldsymbol{x}^*; \boldsymbol{x}']) \geq 0.5$ holds for all solutions, which implies that $S(\boldsymbol{x}^*) = \text{Vol}(\mathcal{X})^{-1} \int_{\mathcal{X}} 1 d\boldsymbol{x}' = 1$. The difficulty in calculating the normalized Copeland score limits its applicability in dueling optimization, thus the soft-Copeland score (González et al., 2017) is adopted, which has the empirically same maximum as the normalized Copeland score. The soft-Copeland score is defined as

$$C(\boldsymbol{x}) = \text{Vol}(\mathcal{X})^{-1} \int_{\mathcal{X}} \pi_f([\boldsymbol{x}; \boldsymbol{x}']) d\boldsymbol{x}'. \tag{3}$$

## 3.2 DIFFERENT SAMPLING RULES IN PREFERENTIAL BAYESIAN OPTIMIZATION.

Rather than classifying dueling optimization methods based on the construction of surrogate models (see Section 2.1), this paper categorizes them according to whether one solution in the next duel is fixed, specifically if the current best solution is used as one of the solution in the next duel.

For methods where one solution in the duel is fixed, such as HB (Takeno et al., 2023) and POP-BO (Xu et al., 2024), the first solution is selected as the current best, while the second solution is resampled based on a given acquisition function. In this case, the pairwise preferences are not entirely independent, as there is a common solution in the duels of consecutive comparisons, which allows a part of relations between different preferences to be inferred. However, this strategy limits the ability of methods to explore the solution space. In contrast, in the second type of methods, both solutions in a candidate duel are resampled through the acquisition functions, with PBO (González et al., 2017) being a typical algorithm of this kind. The PBO uses DTS to choose the potential optimal solution and the most uncertain one for the next duel, thereby balancing exploration and exploitation. However, these approaches lead to pairwise preferences being more isolated, making it challenging to obtain the relations between different preferences.

In this paper, we focus on the second type of methods and aim to uncover the potential preferential relations through a preference propagation technique, thereby enhancing the performance of dueling optimization to match that of function value based methods.

### 3.3 DIRECTED HYPERGRAPH

*Directed hypergraphs* are extension of traditional graphs in which edges, called *directed hyperedges*, can connect multiple vertices from a source set to a target set, unlike traditional graphs where edges only link pairs of vertices. Formally, a directed hypergraph is defined as $\mathcal{G} = (\mathcal{V}, \mathcal{E})$, where $\mathcal{V}$ represents the set of vertices and $\mathcal{E}$ represents the set of directed hyperedges. Each directed hyperedge $\varepsilon \in \mathcal{E}$ is an ordered pair of vertex subsets $(\mathcal{V}_s, \mathcal{V}_t)$, where $\mathcal{V}_s \subseteq \mathcal{V}$ is the source set, and $\mathcal{V}_t \subseteq \mathcal{V}$ is the target set, with $\mathcal{V}_s \cap \mathcal{V}_t = \emptyset$. The directed hyperedge $\varepsilon \in \mathcal{E}$ represents a relationship in which all vertices in the source set $\mathcal{V}_s$ direct to all vertices in the target set $\mathcal{V}_t$. Directed hypergraphs provide a flexible way to model complex interactions between groups of vertices, avoiding the individual connections between each pair, as would be necessary in traditional graphs.

## 4 THE PROPOSED METHOD

Although dueling optimization, e.g., preferential Bayesian optimization, adapts well to scenarios where the objective function can only be evaluated through comparing a pair of solutions, the optimization performance gap still exists between preference based and function value based methods due to the insufficient utilization of pairwise preferences (i.e., which solution is preferred). This section introduces the proposed method, relation augmented preferential Bayesian optimization (RAPBO), which aims to make fuller use of pairwise preferences and enhance the performance of dueling optimization through a preference propagation technique, thereby achieving performance comparable with function value based methods such as Bayesian optimization. To clarify the explanation of the proposed method, we have included a notation section in Appendix E.

### 4.1 RELATION AUGMENTED PREFERENTIAL BAYESIAN OPTIMIZATION

To make fuller use of the pairwise preferences and thus enhance the performance of dueling optimization, the RAPBO method is proposed, with pseudo-code shown in Algorithm 1.

By utilizing a preference propagation technique (detailed in Section 4.2) to make fuller use of the pairwise preferences, and employing PBO as the framework for this process, RAPBO is proposed. The RAPBO begins with an initial dataset $\mathcal{D}_M$, consisting of $M$ evaluated pairwise preferences $\{[\boldsymbol{x}; \boldsymbol{x}'], p\}$, where $p$ indicates whether one solution can beat the other (i.e., 0 for $\boldsymbol{x}'$ and 1 for $\boldsymbol{x}$). In each iteration $j$, RAPBO fits a surrogate model $\mathcal{GP}$ to the current dataset $\mathcal{D}_j$ and performs the preference propagation with parameter $k$ to create an augmented dataset $\mathcal{D}_j^+$ (line 2). This augmented dataset includes additional preferential relations, allowing for fuller utilization of the existing pairwise preferences. A new GP model $\mathcal{GP}^+$ is then trained on $\mathcal{D}_j^+$ to learn the preference function $\pi_{f_p,j}([\boldsymbol{x}; \boldsymbol{x}'])$ (line 3). A sample function $\pi_{\hat{f}_p}$ is drawn from the new GP model $\mathcal{GP}^+$, which guides the selection of the first solution $\boldsymbol{x}_{next}$ (lines 4-5). Next, based on the $\mathcal{GP}$, the solution with the highest uncertainty is chosen as $\boldsymbol{x}'_{next}$ (line 6), resulting in a candidate duel $[\boldsymbol{x}_{next}; \boldsymbol{x}'_{next}]$. Then, the duel is evaluated, and the resulting preference $p_{j+1}$ is used to update the dataset to $\mathcal{D}_{j+1}$ (lines 7-8). It is worth noting that the additional preferential relations generated by the preference

---

**Algorithm 1** Relation Augmented Preferential Bayesian Optimization (RAPBO)

---

**Input:** Initial dataset $\mathcal{D}_M = \{[\boldsymbol{x}_i; \boldsymbol{x}_i'], p_i\}_{i=1}^M$, number of available duels $N$, boundary of subspace $\mathcal{X} \subset \mathbb{R}^D$ and preference propagation parameter $k$.

**Procedure:**
1: **for** $j = M$ **to** $M + N - 1$ **do**
2:     Fit a $\mathcal{GP}$ to $\mathcal{D}_j$ and perform preference propagation with parameter $k$ to obtain the augmented dataset $\mathcal{D}_j^+$.
3:     Fit a $\mathcal{GP}^+$ to $\mathcal{D}_j^+$ and learn $\pi_{f_p,j}([\boldsymbol{x}; \boldsymbol{x}'])$.
4:     Sample a function $\pi_{\hat{f}_p}$ from $\mathcal{GP}^+$.
5:     $\boldsymbol{x}_{next} = \text{argmax}_{\boldsymbol{x} \in \mathcal{X}} \int_{\mathcal{X}} \pi_{\hat{f}_p}([\boldsymbol{x}; \boldsymbol{x}']; \mathcal{D}_j^+) \mathrm{d}\boldsymbol{x}'$ .
6:     $\boldsymbol{x}_{next}' = \text{argmax}_{\boldsymbol{x}' \in \mathcal{X}} \sigma(\mathcal{GP}|\boldsymbol{x} = \boldsymbol{x}_{next}, \mathcal{D}_j)$ .
7:     Run the duel $[\boldsymbol{x}_{next}; \boldsymbol{x}_{next}']$ and obtain $p_{j+1}$.
8:     Augment $\mathcal{D}_{j+1} = \{\mathcal{D}_j \cup ([\boldsymbol{x}_{next}; \boldsymbol{x}_{next}'], p_{j+1})\}$.
9: **end for**
10: Fit a $\mathcal{GP}$ to $\mathcal{D}_{M+N}$ and find the solution $\boldsymbol{x}^*$ with the highest soft-Copeland score.
11: **return** $\boldsymbol{x}^*$.

---

propagation technique in each iteration do not carry over to the next iteration. After $N$ iterations, the final GP model is fit to the complete dataset $\mathcal{D}_{M+N}$, and the optimal solution $\boldsymbol{x}^*$ is determined based on the highest soft-Copeland score (line 10).

In the following sections, we will provide a detailed explanation of the preference propagation technique as well as the time and space complexity of the technique.

### 4.2 PREFERENCE PROPAGATION TECHNIQUE

In order to make fuller use of the pairwise preferences, a preference propagation technique is used to uncover potential relations between different preferences, with the pseudo-code detailed in Appendix C. The preference propagation technique first clusters solutions using a clustering algorithm. Specifically, we employ a Gaussian mixture model (Reynolds et al., 2009), which excels at capturing complex data distributions by modeling them as a combination of multiple Gaussian components. After identifying the solution set with the highest intra-cluster similarity, the technique utilizes a hypergraph to model the relations between solutions, achieving relation augmentation. This technique enables a fuller utilization of the pairwise preferences, ultimately enhancing the optimization process.

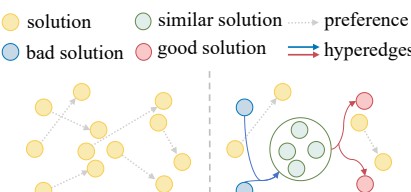

Preference Propagation Technique

Figure 1: A diagram of the preference propagation technique. The pairwise preferences are modeled as a directed graph, where each solution is represented by a vertex and each preference is represented by a directed edge, pointing from the worse solution to the better solution (left). And after preference propagating, a directed hypergraph is used to model the relations between the solutions (right).

Inspired by Sui et al. (2017), we model a special Gaussian process $\mathcal{GP}^D$, where the kernel is initially set as $1.0 * RBF(1.0)$, to fit the function where the value represents the probability of one solution beating the optimal solution, ensuring that $\mathcal{GP}^D$ operates in the $D$-dimensional solution space. Then, $\mathcal{GP}^D$ can be used to compute the covariance between any two solutions in the dataset, which can serve as a measure of similarity between the two solutions. Finally, these similarities will be transformed into distances, specifically $1 - similarity$, and clustering will be performed based on these distances, resulting in a set of solutions with the highest intra-cluster similarity (i.e., the smallest intra-cluster distance).

As Figure 1 shown, pairwise preferences are modeled as a directed graph, where each vertex represents a solution, and each preference corresponds to a directed edge pointing from the worse solution to the better solution. Next, preference propagation is conducted on the current dataset, with the set of all solutions defined as $\mathcal{V}$. The preference propagation technique first utilizes clustering based on

the surrogate model $\mathcal{GP}^D$ to partition all solutions into $k$ clusters and obtain a solution set with the highest intra-cluster similarity (the green circle), where the solutions in this set are termed *similar solutions* (the green vertices), and this set is defined as $\mathcal{V}_s \subseteq \mathcal{V}$. We assume that similar solutions exhibit analogous relations, meaning that if A and B are similar solutions and A is preferred over C, then B is also preferred over C. Subsequently, all solutions that can direct towards similar solutions via directed edges are termed *bad solutions* (the blue vertices), forming the set of the bad solutions $\mathcal{V}_{\mathrm{bad}} \subseteq \mathcal{V}$, while all solutions that can be reached from similar solutions through directed edges are termed *good solutions* (the red vertices), forming the set of the good solutions $\mathcal{V}_{\mathrm{good}} \subseteq \mathcal{V}$. The sets $\mathcal{V}_{\mathrm{bad}}$, $\mathcal{V}_s$, and $\mathcal{V}_{\mathrm{good}}$ have no intersection with each other. Finally, we construct a complete directed hypergraph $\mathcal{G}$ using two directed hyperedges. Specifically, $\varepsilon_1$ directs from the set of bad solutions to the set of similar solutions, i.e., $\varepsilon_1$ is an ordered pair of sets $(\mathcal{V}_{\mathrm{bad}}, \mathcal{V}_s)$, and $\varepsilon_2$ directs from the set of similar solutions to the set of good solutions, i.e., $\varepsilon_2$ is an ordered pair of sets $(\mathcal{V}_s, \mathcal{V}_{\mathrm{good}})$.

Based on this directed hypergraph $\mathcal{G} = (\mathcal{V}, \mathcal{E})$, where $\mathcal{E} = \{\varepsilon_1, \varepsilon_2\}$, RAPBO can uncover more potential preferential relations, i.e., all similar solutions are better than the bad solutions, and all good solutions are better than the similar solutions. Moreover, by leveraging the transitivity of preferences, we can also conclude that all good solutions are better than the bad solutions. Thus, the preference propagation technique realizes relation augmentation based on the existing dataset, enabling a fuller utilization of the pairwise preferences.

### 4.3 COMPLEXITY ANALYSIS

In this section, we analyze the improvements in time and space complexity achieved by using hypergraphs to model the relations between solutions in the preference propagation technique.

The introduction of hypergraphs avoids the full connection that occurs when traditional graphs are used in the preference propagation technique. To establish the connections between the three solution sets, a traditional graph requires full connections from the bad solution set to the similar solution set, and from the similar solution set to the good solution set. We denote the quantities of bad solutions, similar solutions, and good solutions as $n_1$, $n_2$ and $n_3$, respectively. Specifically, in the case of using a traditional graph, the time complexity of modeling the relations between solutions is $O(n_1 * n_2 + n_2 * n_3)$, and the space complexity of the preference propagation technique is also $O(n_1 * n_2 + n_2 * n_3)$. However, when employing a hypergraph instead of a traditional graph, the two solution sets can be directly connected through a single hyperedge, resulting in the time complexity of modeling the relations reducing to $O(m)$, where $m$ is the number of hyperedges and $m = 2$ in the preference propagation technique. Thus, the time complexity can also be expressed as $O(2)$. Additionally, the space complexity of the preference propagation technique also decreases to $O(m + n_1 + n_2 + n_3)$ with $m = 2$. The reduction in complexity brought about by the directed hypergraphs makes the preference propagation technique more efficient and practical.

## 5 EXPERIMENT

In this section, we compare RAPBO with a series of dueling optimization algorithms through experiments on synthetic functions and real-world tasks. RAPBO is implemented by BoTorch (Balandat et al., 2020) and our experimental codes are publicly available at `https://anonymous.4open.science/r/RAPBO-E15F`. RAPBO uses a Gaussian process with default parameters from the BoTorch library as the surrogate model, and employs CMA-ES (Hansen et al., 2003) as the optimizer of the acquisition function. We compare RAPBO with four dueling optimization methods, where both solutions in a candidate duel are resampled based on specific acquisition functions, rather than having one solution fixed as the current best, such as HB (Takeno et al., 2023) and POP-BO (Xu et al., 2024). The methods include PBO (González et al., 2017), KSS (Sui et al., 2017), qEUBO (Astudillo et al., 2023) and a simplified version of COMP-UCB (Xu et al., 2020), which omits the second part of the optimization process that depends on function values. Specifically, PBO can be regarded as the version of RAPBO after ablating the preference propagation technique. The experiments are designed to answer the following four significant questions.

Q1: Effectiveness and superiority: Can RAPBO handle dueling optimization tasks and achieve better performance than other dueling optimization methods?

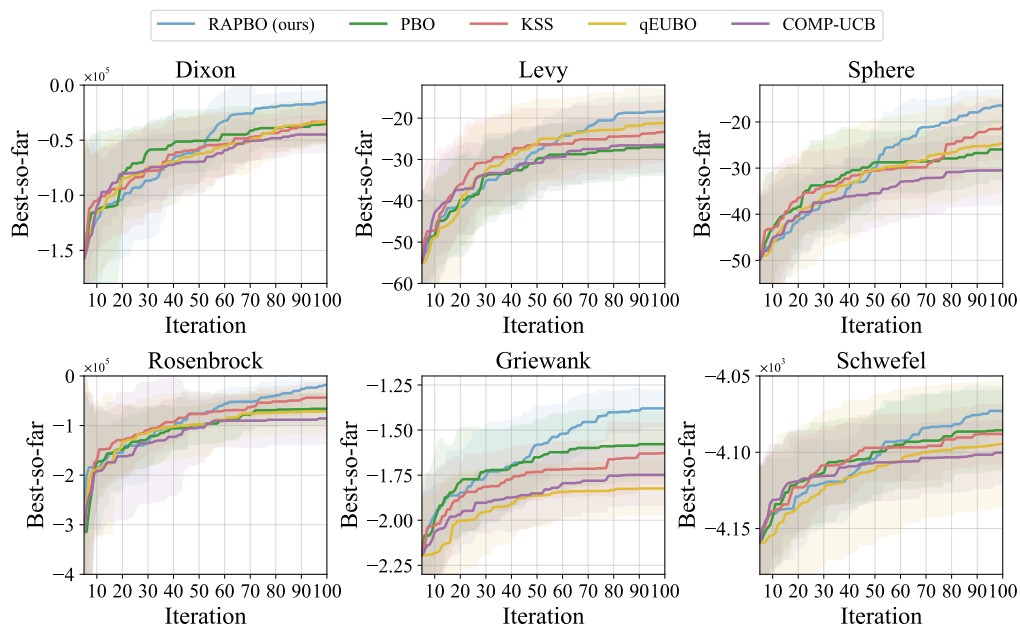

Figure 2: The best function value found by RAPBO on synthetic functions are compared with different dueling optimization algorithms. All methods are evaluated with 5 initial duels, 100 iterations, and each experiment is repeated 20 times. The mean and standard deviation of the results are plotted. The horizontal axis of the plots represents the number of evaluations, and the vertical axis represents the best function value found by the algorithm.

Q2: Utilization: Dose RAPBO uncover potential preferential relations based on the existing preferences and make fuller use of pairwise preferences?

Q3: The benefit of dueling optimization: Under a fixed budget, can RAPBO match or even surpass the performance of function value based Bayesian optimization methods?

Q4: The impact of hyper-parameters: How sensitive is RAPBO to changes in hyper-parameters?

The four questions are answered sequentially in this section. For all tasks, the best function value found so far is used as the evaluation criterion.

## 5.1 EXPERIMENTAL SETTINGS

**The Setting of Synthetic Functions.** To evaluate the performance of RAPBO, experiments are first conducted on synthetic functions. In this paper, we construct objective functions for evaluation in a standard setting based on different synthetic functions[1]. Specifically, let $f : \mathbb{R}^D \to \mathbb{R}$ be a base synthetic function, with its domain adjusted to $[-1, 1]^D$. The input is an $D$-dimensional vector $\boldsymbol{x} = [x_1, x_2, \ldots, x_D]$, and the output is the function value $f(\boldsymbol{x})$ for this input. In the experiments, we evaluate RAPBO on six synthetic functions with $D = 10$, namely Dixon-Price, Levy, Sphere, Rosenbrock, Griewank, and Schwefel. These synthetic functions collectively cover various optimization problem types, including multimodal landscapes, complex terrains, periodic variations, and convex optimization. All experiments on synthetic functions are maximization optimization.

**The Setting of Real-world Tasks.** To further explore the performance of RAPBO and its applicability to real-world tasks, RAPBO is evaluated on three real-world datasets. The first dataset is RobotPush problem (Eriksson et al., 2019), which is a noisy 14-dimensional motion control problem involving optimizing the pre-image for pushing an object to a goal location. The second dataset is Sagas (Schlueter et al., 2021), a 12-dimensional problem, which is designed for trajectory optimization problems, aiming to minimize the overall mission length to reach targets. The third dataset is a 10-dimensional problem, Cassini1-MINLP (Schlueter & Munetomo, 2019), which is designed to

---

[1]http://www.sfu.ca/~ssurjano/optimization.html

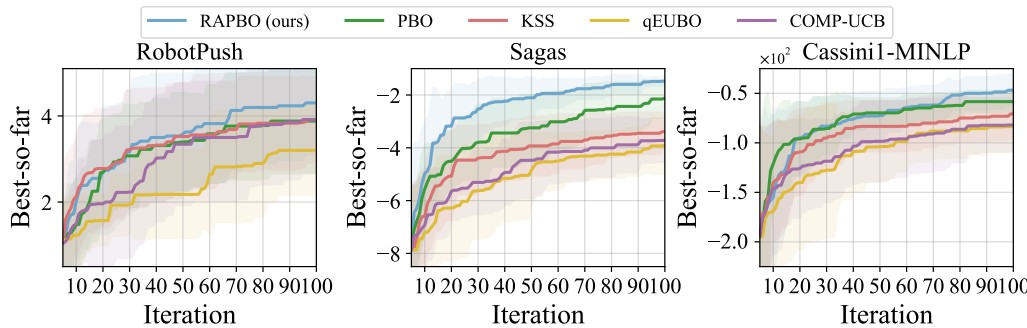

Figure 3: The best function value found by algorithms on real-world datasets. Each experiment is repeated 20 times. The mean and standard deviation of the results are plotted. The horizontal axis of the plots represents the number of evaluations, and the vertical axis represents the best function value found by the algorithm. All methods are evaluated with 5 initial duels and 100 iterations.

optimize a mixed-integer nonlinear programming problem (MINLP), allowing for flexible selection of any planet in the solar system. These real-world datasets are well-suited for dueling optimization. RobotPush is a noisy dataset where the noise affects the performance of function value based methods, while dueling optimization can mitigate the impact of noise to some extent. Cassini1-MINLP and Sagas are spacecraft trajectory optimization problems where evaluating the function value of a given solution may be very costly and time-consuming, while comparing a pair of solutions is much more manageable. All real-world tasks are maximization tasks.

## 5.2 THE PERFORMANCE OF RAPBO

**About Q1: Effectiveness and Superiority.** In the synthetic functions and real-world tasks experiments, $I = 500$ samples are employed to estimate the integral of the soft-Copeland score, and the GP model is initialized using $M = 5$ duels, followed by $N = 95$ duels for the optimization process. For RAPBO, we use $k = 3$ to execute the preference propagation technique. For more detailed algorithm parameter settings, refer to the Appendix C. All experiments are repeated 20 times and the results are shown in Figure 2 and 3. More detailed results are in the Appendix D.

Across all synthetic functions, RAPBO consistently achieves better performance compared to the other optimization methods, showcasing its ability to handle dueling optimization tasks well. The RAPBO curve converges relatively quickly and remains below other methods at around 50 iterations, indicating that it finds better solutions earlier in the optimization process. Moreover, RAPBO shows a stable improvement in performance during optimization, particularly as other methods begin to converge around iterations 70 (a phenomenon we will explore further in the next section). Finally, the standard deviation of RAPBO is relatively narrow in most cases, suggesting that its performance is more reliable compared to the other methods, particularly in challenging functions like Griewank.

Across all real-world tasks, the RAPBO also achieves the best results. In RobotPush task, PBO, KSS, and COMP-UCB all achieve the similar final performance, as they are troubled by noise during optimization. However, due to the preference propagation technique, which uncovers many potential preferential relations from the existing preferences, RAPBO can find the better solutions. In Sagas and Cassini1-MINLP tasks, RAPBO exhibits a stable improvement throughout the optimization process, and ultimately achieve the best results.

In a nutshell, the experimental results verify that RAPBO can handle dueling optimization tasks well and reflect the superiority of RAPBO over other dueling optimization methods, which answers Q1.

**About Q2: Utilization.** To explore the utilization of pairwise preferences in RAPBO and explain why RAPBO shows a stable improvement in performance, we analyze the augmented preferences to better understand the factors driving the algorithm performance, as shown in Figure 4. The experiments conduct on the Griewank function and three real-world tasks, with all settings consistent with those in the above section, and the experiments are repeated 20 times.

As shown in Figure 4, a significant number of augmented preferences are newly added after preference propagation, and these preferences all maintain a high accuracy, which verifies that pairwise preferences are not fully utilized in previous work like PBO (González et al., 2017).

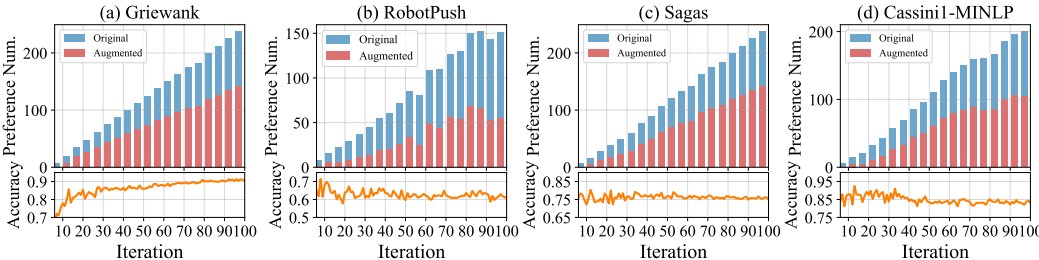

Figure 4: The utilization of RAPBO on Griewank function and three real-world tasks. The figure shows the mean number of preferences in the original dataset (blue) and the augmented preferences newly added after preference propagation (red) in the top plot, as well as the mean accuracy of the augmented preferences in the bottom plot. During the optimization process, RAPBO uses a combination of original preferences and augmented preferences (blue + red). All settings are the same as Figure 2 and Figure 3, and each experiment is repeated 20 times.

The Figure 4(a) illustrates the results on the Griewank function, which we consider as an ideal environment. In the top plot, it is clear that as optimization progresses, the number of newly added augmented preferences significantly exceeds that of original preferences, with a faster growth rate as well. The bottom plot shows the mean accuracy of the augmented preferences, which increases steadily throughout the optimization process, consistently remaining above 0.5. Additionally, the lower accuracy of the augmented preferences during the early process of optimization may explain why RAPBO performs worse than methods like PBO and KSS in certain situations, as shown in Figure 2, and as the accuracy of the augmented preferences increases, the performance of RAPBO also improves rapidly. The Figure 4(b) shows the results on the RobotPush task, and due to the presence of the noise, the accuracy of the augmented preferences is relatively low, but it remains consistently above 0.5. In this context, the preference propagation technique does not merely seek to propagating more relations, but instead uncovers a limited number of relations from the existing pairwise preferences, i.e., the scope of preference propagation is relatively narrow. This behavior ensures that the accuracy of the augmented preferences does not decline further, thereby preventing the newly generated preferential relations from affecting optimization performance. The Figure 4(c) and (d) show the results on the Sagas and Cassini1-MINLP tasks, respectively. In both tasks, the augmented preferences all exhibit relatively high accuracy, which encourages the preference propagation technique to uncover more preferential relations from the existing dataset, i.e., the scope of preference propagation is relatively broad. In the three real-world tasks, due to the complexity of the tasks, there is no gradual increase in accuracy of the augmented preferences as shown in Figure 4(a).

In a nutshell, the results indicate that RAPBO has effectively uncovered the potential preferential relations, thereby further utilizing the available preferences, which answers Q2.

## 5.3 Dueling Optimization vs. Function Value based Optimization

**About Q3: Benefit of Dueling Optimization.** To explore the optimization performance gap between preference based and function value based methods, and verify that the performance of RAPBO can match or even surpass that of function value based Bayesian optimization methods, RAPBO and PBO (regarded as the ablated version of RAPBO) are compared with the function value based method, GP-UCB (Srinivas et al., 2010). All methods are tested on the real-world tasks and repeated 20 times, with the results shown in Figure 5. In Figure 5(a), (b) and (c), the cost of evaluating the function value is set to be twice expensive as that of comparing a pair of solutions, and GP-UCB is initialized with 15 random solutions for a better initialization. While in Figure 5(d), it is set to be 1.5 times as expensive and GP-UCB is initialized with 20 solutions. In all experiments, RAPBO and PBO is initialized with $M = 30$ random duels. In Figure 5(a), (b) and (c), all methods have a budget of 100 while in Figure 5(d), the budget is set to 90.

In Figure 5(a), (b) and (c), the initial value of the function value based method is found to be worse than that of the two other preference based methods after initialization. This is because the function value based method only randomly selects 15 solutions from the solution space for initialization, while the preference based methods randomly select 60 solutions from the solution space, which are then paired into 30 duels for initialization. However, due to the more informative solution evalua-

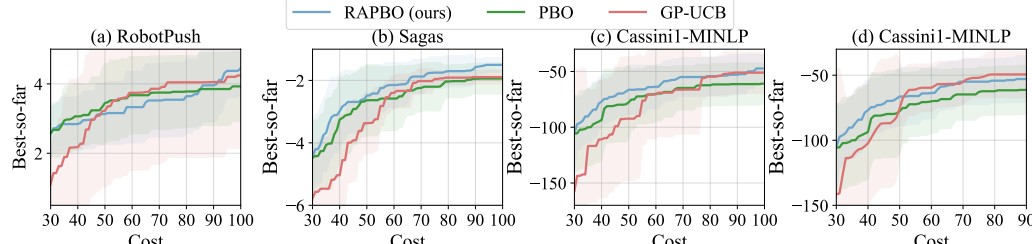

Figure 5: The best function value found by algorithms with fixed budget. In (a), (b) and (c), evaluating the function value is twice as expensive as comparing a pair of solutions, and in (d) it is set to be 1.5 times as expensive. Each experiment is repeated 20 times. The mean and standard deviation of the result are plotted. The vertical axis represents the best function value found by the algorithm and the horizontal axis of the plots represents is the cost that the algorithm has used.

tions, GP-UCB shows a rapid improvement in performance, subsequently surpassing that of PBO, which exhibits *a clear optimization performance gap between preference based and function value based methods*. In the RobotPush and Cassini1-MINLP tasks, RAPBO continues to exhibit a stable improvement, ultimately achieving performance comparable to those of GP-UCB. However, in Sagas task, due to the advantages of pairwise preferences, RAPBO consistently outperforms GP-UCB while continuously improving. To further verify that the performance of RAPBO under a fixed budget can match that of value based methods, we conduct additional experiment on Cassini1-MINLP task and set the cost of evaluating the function value to be 1.5 times that of comparing a pair of solutions, as shown in Figure 5(d). It can be found that the performance of GP-UCB quickly surpasses that of RAPBO, but RAPBO shows a stable improvement in performance and achieves performance similar to that of GP-UCB when the cost is exhausted.

In a nutshell, these results verify that, ***under the same cost budget, RAPBO is competitive with or even surpass the function value based Bayesian optimization methods with respect to optimization performance***. It for the first time indicates that, if preferential relations between solutions within different preferences are fully and deeply exploited and utilized, dueling optimization could be more effective for expensive and costly optimization tasks, which answers Q3.

### 5.4 HYPER-PARAMETER ANALYSIS

**About Q4: Impact of Hyper-parameters.** To explore the sensitivity of RAPBO to different hyper-parameters, we conduct hyper-parameter experiments for $k$ on all synthetic functions, with the results shown in Appendix B. It can be found that RAPBO consistently outperforms PBO (regarded as the ablated version of RAPBO) across different hyper-parameter $k$ and is not significantly affected by changes in $k$, showcasing its insensitivity to hyper-parameter variations, which answers Q4. Additionally, RAPBO consistently showcases a stable improvement in performance, indicating that the preference propagation technique still operates reliably across all hyper-parameters $k$.

## 6 CONCLUSION AND DISCUSSION

This paper aims to fill the optimization performance gap between preference based and function value based methods, and verify that the preference based methods can match or even surpass the performance of the function valued based methods. We propose the method, relation augmented preferential Bayesian optimization (RAPBO), which enhances the performance of dueling optimization by capturing potential preferential relations through the proposed preference propagation technique. Extensive experiments on synthetic functions and real-world tasks disclose the satisfactory accuracy of augmented preferences in RAPBO, and exhibit the superiority of RAPBO compared with existing dueling optimization methods. Notably, it is verified that the performance of RAPBO can match or even surpass that of the function value based Bayesian optimization methods under the same cost budget. In future work, we plan to utilize the pairwise preferences more fully through more efficient methods, and further improve the accuracy of the augmented preferences to enhance the performance of dueling optimization.

## 7 ETHICS AND REPRODUCIBILITY STATEMENTS

**Ethics.** This work does not include any human subjects, personal data, or sensitive information. All testing datasets utilized are publicly accessible, and no proprietary or confidential information has been employed.

**Reproducibility.** Experimental settings are described in Section 5.1 with further details of the methods included in Appendix C. The datasets utilized in this paper are all publicly available and open-source. The link to our anonymous code repository is `https://anonymous.4open.science/r/RAPBO-E15F`.

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

APPENDIX

## A  THE PSEUDO-CODE OF THE PREFERENCE PROPAGATION TECHNIQUE

---

**Algorithm 2** Preference Propagation Technique

---

**Input:** Current dataset $\mathcal{D}_j = \{[\boldsymbol{x}_i; \boldsymbol{x}_i'], p_i\}_{i=1}^j$, and preference propagation parameter $k$.
**Procedure:**
1: Fit a special $\mathcal{GP}^D$ to $\mathcal{D}_j$ and compute the covariance between any two solutions in the dataset $\mathcal{D}_j$ to assess their similarity.
2: Compute distance between any two solutions by $1 - similarity$, and cluster all solutions into $k$ sets.
3: Obtain the set of similar solutions $\mathcal{V}_s$ with the highest intra-cluster similarity, the set of bad solutions $\mathcal{V}_{\text{bad}}$ and the set of good solutions $\mathcal{V}_{\text{good}}$.
4: Construct the directed hyperedges $\varepsilon_1$ and $\varepsilon_2$ to model the potential preferential relations.
5: Combine the augmented preferential relations with the dataset $\mathcal{D}_j$ and obtain the augmented dataset $\mathcal{D}_j^+$.
6: **return** the augmented dataset $\mathcal{D}_j^+$.

---

The preference propagation technique, as shown in Algorithm 2, is designed to make fuller utilization of the existing pairwise preferences by modeling potential preferential relations among solutions. Initially, it requires the current dataset $\mathcal{D}_j$ and a preference propagation parameter $k$. The preference propagation technique begins by fitting a specific Gaussian process model $\mathcal{GP}^D$ to the dataset $\mathcal{D}_j$ and computing the covariance between solutions to assess their similarity (line 1). Next, the technique calculates distances based on the complement of similarity and clusters the solutions into $k$ sets (line 2). From these clusters, it identifies a set of similar solutions $\mathcal{V}_s$ with the highest intra-cluster similarity, as well as the set of bad solutions $\mathcal{V}_{\text{bad}}$ and the set of good solutions $\mathcal{V}_{\text{good}}$ (line 3). Directed hyperedges are constructed to model the potential preferential relations among these solution sets (line 4). Finally, the augmented preferential relations are combined with the original dataset $\mathcal{D}_j$ to create an augmented dataset $\mathcal{D}_j^+$ (line 5), which is then returned as output. This technique aims to better uncover and utilize the potential preferential relations between preferences, thereby make fuller utilization of the existing pairwise preferences.

## B  HYPER-PARAMETER ANALYSIS

The hyper-parameter analysis experiments for $k$ are conducted on all synthetic functions, with the results shown in Figure 6. In the experiments, $I = 500$ samples are employed to estimate the integral of the soft-Copeland score, and the GP model is initialized using $M = 5$ duels, followed by $N = 95$ duels for the optimization process. For RAPBO, a series of hyper-parameter values for $k$ are used to execute the preference propagation technique. For comparison, the final results of PBO, regarded as a version of RAPBO after ablating the preference propagation technique, are also plotted. The results clearly show that RAPBO consistently surpasses PBO across various hyper-parameter values of $k$, highlighting its insensitivity to changes in $k$. This characteristic ensures the adaptability and stability of RAPBO across different application scenarios. Furthermore, regardless of the hyper-parameter $k$, RAPBO consistently shows a stable improvements in performance, which indicates that the preference propagation technique operates reliably, showcasing its reliability and consistency under varying conditions.

To further analyze why RAPBO is not sensitive to changes in the hyper-parameter $k$, we explore the behavior of the preference propagation technique under different values of $k$ on the Griewank function. Figure 7 shows the mean number of original preferences at the beginning of each iteration and the newly added augmented preferences after preference propagation (top), as well as the mean accuracy of the augmented preferences (bottom). From the figure, we observe that within a limited range, the choice of the hyper-parameter $k$ does not significantly affect the number of new augmented preferences added after preference propagation, nor their accuracy during the optimization process. Therefore, the hyper-parameter $k$ do not significantly affect the relation augmentation

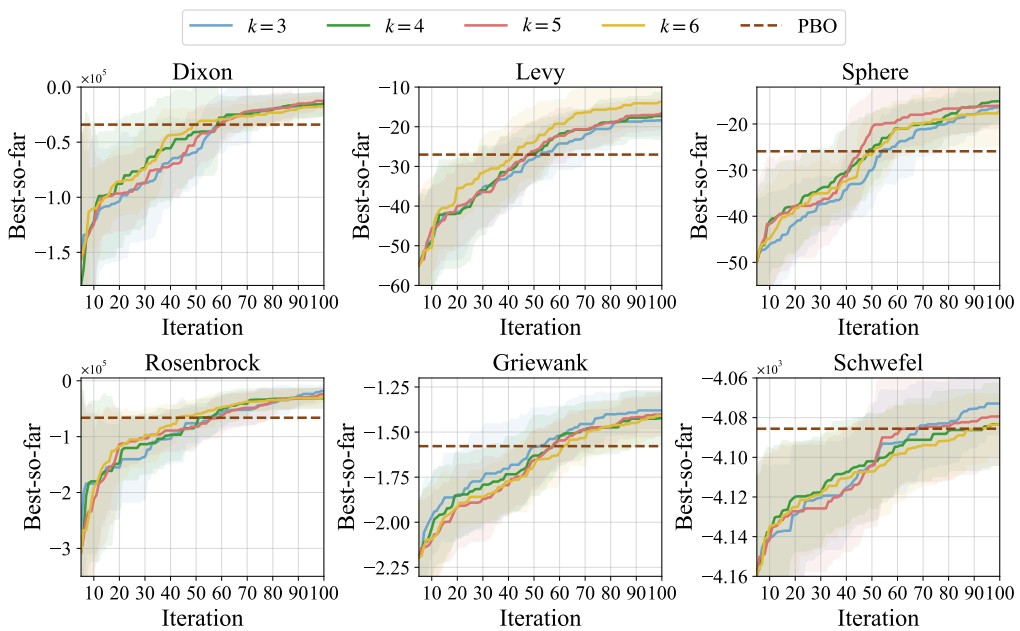

Figure 6: Hyper-parameter analysis on synthetic functions. Each experiment is repeated 20 times and the final results of PBO are also plotted. The mean and standard deviation of the best function value found are plotted. The horizontal axis of the plots represents the number of evaluations, and the vertical axis represents the best function value found by the algorithm.

effect of the preference propagation technique on the existing dataset, allowing RAPBO to achieve better optimization performance. In fact, within the preference propagation technique, after GMM performs clustering, we only select the solutions from the cluster with the highest intra-cluster similarity as the similar solutions, and the role of GMM is to help us select the most similar batch of solutions. Therefore, the choice of $k$ does not significantly affect the performance of RAPBO.

These results explain why the optimization performance of RAPBO is not sensitive to changes in the hyper-parameter $k$ and it further verifies that the preference propagation technique can run stably under different conditions.

## C  IMPLEMENTATION DETAILS OF OPTIMIZATION METHODS

**PBO** (González et al., 2017): PBO, the first framework to extend Bayesian optimization to scenarios where only information about user preferences can be obtained, is repeated using the BoTorch framework in the experiments and follows the same hyper-parameter specifications as outlined in Zhang et al. (2023).

**KSS** (Sui et al., 2017): KSS is an algorithm that effectively addresses the multi-dueling bandits problem by reducing it to a conventional bandit setting, and it can also be applied to dueling optimization. We use the code from the GitHub repository: `https://github.com/Zhangywh/PE-DBO`.

**COMP-UCB** (Xu et al., 2020): COMP-UCB is the simplified version that omits the second part of the optimization process that depends on function values. We use the code from the GitHub repository: `https://github.com/Zhangywh/PE-DBO`.

**qEUBO** (Astudillo et al., 2023): qEUBO provides a promising acquisition function with a grounded decision-theoretic justification. We use the implementation from the author's GitHub repository: `https://github.com/RaulAstudillo06/qEUBO`.

**GP-UCB** Srinivas et al. (2010): GP-UCB is a Bayesian optimization algorithm with the upper confidence bound strategy that builds a model to predict an unknown function, balancing exploration

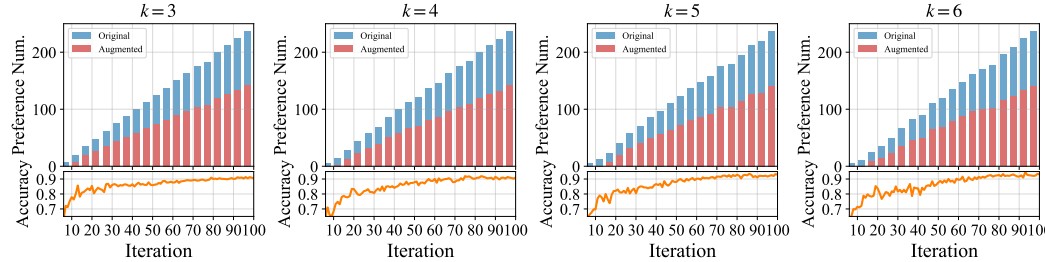

Figure 7: Investigating the behavior of the preference propagation technique under different hyperparameter values of $k$ on the Griewank function ($D = 10$). The figure shows the mean number of preferences in the original dataset (blue) and the augmented preferences newly added after preference propagation (red) in the top plot, as well as the mean accuracy of the augmented preferences in the bottom plot. All settings are the same as Figure 6, and each experiment is repeated 20 times.

and exploitation. In the experiments, the BoTorch framework is used to implement GP-UCB, with $\beta$ defined as $0.2D\log(2n)$, where $D$ is the dimension of the solution space and $n$ is the number of samples in the dataset.

# D  DETAILED RESULTS

Table 1 and Table 2 record the final mean convergence value of various algorithms under each experimental environment. In order to verify that RAPBO statistically outperforms other baselines in most cases, we perform t-tests with a significance level of $0.05$. As shown in the tables, in most tasks, RAPBO statistically outperforms other dueling optimization methods. The results show that RAPBO can handle dueling optimization tasks well and reflect the superiority of RAPBO over other dueling optimization methods.

Table 1: The detailed results of dueling optimization methods on synthetic functions. In each column, an entry with the best mean value is marked in bold and underline for the runner-up. If the mean value of the best method significantly differs from the runner-up, passing a t-test with a significance level of $0.05$, then we denote it with "*" at the corresponding position.

| Method | Rosenbrock | Dixon | Griewank | Levy | Schwefel | Sphere |
|---|---|---|---|---|---|---|
| PBO | $-66175.670 \pm 31607.440$ | $-34068.125 \pm 18046.893$ | $-1.578 \pm 0.180$ | $-27.021 \pm 6.795$ | $-4085.577 \pm 25.830$ | $-25.920 \pm 6.396$ |
| KSS | $-43576.960 \pm 30242.922$ | $-32522.400 \pm 17915.740$ | $-1.624 \pm 0.168$ | $-23.338 \pm 7.265$ | $-4088.018 \pm 19.577$ | $-21.097 \pm 6.080$ |
| qEUBO | $-71249.730 \pm 45543.810$ | $-33604.113 \pm 18374.818$ | $-1.824 \pm 0.147$ | $-21.008 \pm 8.093$ | $-4094.577 \pm 41.635$ | $-24.646 \pm 10.413$ |
| COMP-UCB | $-85650.450 \pm 51355.145$ | $-44837.203 \pm 16389.234$ | $-1.748 \pm 0.168$ | $-26.435 \pm 6.614$ | $-4100.247 \pm 20.779$ | $-30.497 \pm 7.499$ |
| RAPBO | $\mathbf{-18416.717 \pm 11828.153}$* | $\mathbf{-15456.545 \pm 10959.241}$* | $\mathbf{-1.3798 \pm 0.100}$* | $\mathbf{-18.388 \pm 3.880}$ | $\mathbf{-4072.887 \pm 15.641}$ | $\mathbf{-16.432 \pm 3.480}$* |

Table 2: The detailed results of dueling optimization methods on real-world datasets. In each column, an entry with the best mean value is marked in bold and underline for the runner-up. If the mean value of the best method significantly differs from the runner-up, passing a t-test with a significance level of $0.05$, then we denote it with "*" at the corresponding position.

| Method | RobotPush | Cassini1-MINLP | Sagas |
|---|---|---|---|
| PBO | $3.881 \pm 1.221$ | $-58.359 \pm 11.105$ | $-2.117 \pm 0.730$ |
| KSS | $3.909 \pm 1.000$ | $-70.781 \pm 25.648$ | $-3.382 \pm 0.677$ |
| qEUBO | $3.204 \pm 1.038$ | $-83.197 \pm 25.342$ | $-3.922 \pm 1.071$ |
| COMP-UCB | $3.918 \pm 1.247$ | $-81.956 \pm 27.205$ | $-3.721 \pm 0.825$ |
| RAPBO | $\mathbf{4.302 \pm 1.205}$ | $\mathbf{-46.850 \pm 16.175}$* | $\mathbf{-1.478 \pm 0.169}$* |

# E  NOTATION FOR THE PROPOSED METHOD

In order to facilitate a better understanding of the proposed method, we present the notation used throughout this paper. Table 3 summarizes the key symbols and their corresponding meanings, providing clarity on the mathematical components and variables involved in our approach, with all other symbols derived from those in the table.

Table 3: Notation for the proposed method.

| Symbol | Meaning | Symbol | Meaning |
|--------|---------|--------|---------|
| $\mathcal{X}$ | Solution space | $\mathcal{D}$ | Dataset |
| $\boldsymbol{x}$ | Solution | $[\boldsymbol{x}, \boldsymbol{x}']$ | Duel |
| $p$ | Preference | $\pi_{f_p}$ | Preference function |
| $\mathcal{G}$ | Directed hypergraph | $\mathcal{V}$ | A set of vertices |
| $\varepsilon$ | Directed hyperedge | $\mathcal{E}$ | A set of directed hyperedges |
| $k$ | Preference propagation parameter | $\mathcal{GP}$ | Gaussian process |
| $I$ | Number of iterations | $M$ | Number of initial solutions |
| $N$ | Number of duels | $D$ | Dimension of the solution space |

