# OpenReview forum: "Relation Augmented Preferential Bayesian Optimization via Preference Propagation"
_ICLR.cc/2025/Conference — ICLR 2025 Conference Withdrawn Submission_

### Official Review · Reviewer_VX5p · 2024-10-27

**Soundness:** 3
**Presentation:** 2
**Contribution:** 2
**Rating:** 6
**Confidence:** 2

**Summary:**

In this paper, the authors propose relation augmented preferential Bayesian optimization (RAPBO) via preference propagation. RAPBO aims to uncover the potential preferential relations between solutions within different preferences through the proposed preferential relation propagation technique. The experimental results disclose the satisfactory accuracy of augmented preferences in RAPBO, and show the superiority of RAPBO compared with existing dueling optimization methods.

**Strengths:**

S1: The paper is easy to follow and well-organized.

S2: The authors open source the code for reproduction.

S3: The authors conduct extensive experiments to validate the efficacy of the proposed method.

**Weaknesses:**

W1: The font in Fig.1 is too small. Please kindly enlarge the font in Fig.1 to make it more readable.

W2: The authors should provide a notation in the methodology section.

**Questions:**

NA

---

> ### Author Response · Authors · 2024-11-17
> **Reply to Reviewer VX5p**
>
> We greatly appreciate your thorough review and valuable comments. We have carefully addressed each of your concerns as detailed below.
>
> **Q1: The font in Fig.1 is too small. Please kindly enlarge the font in Fig.1 to make it more readable.**
>
> Thank you for your helpful comment.
>
> We revise the layout and redrawing of all figures to improve their readability, including increasing the font size and ensuring that the overall design of the figures enhances clarity. Additionally, we will make the figure descriptions more concise and clear to further improve their accessibility. We appreciate your suggestion and will make sure that the figures are more readable and effectively communicated in the revised version.
>
> **Q2: The authors should provide a notation in the methodology section.**
>
> Thank you for your constructive feedback.
>
> The inclusion of a notation in the methodology section will certainly make our proposed approach easier to understand. Our main notation is as shown in the table below, with all other symbols derived from those in the table.
>
> Table1: Notation in the methodology section.
>
> | **Symbol**  | **Meaning**|**Symbol**  | **Meaning**|
> |:----------:|:----------:|:----------:|:------------------:|
> | $ \mathcal{X} $   | Solution space |  $\mathcal{D}$  |  Dataset     |
> | $ \boldsymbol{x} $   | Solution  |     $[\boldsymbol{x}, \boldsymbol{x}']$       |     Duel         |
> | $ p $ | Preference       |       $ \pi_{f_p} $       |    Preference function       |
> | $\mathcal{G}$ | Directed hypergraph | $\mathcal{V}$ | A set of vertices |
> | $ \varepsilon $ | Directed hyperedge | $ \mathcal{E}$ | A set of directed hyperedges |
> | $k$ | Preference propagation parameter | $\mathcal{GP}$| Gaussian process |
> | $I$|Number of iterations | $M$ | Number of initial solutions |
> | $N$| Number of duels |$D$ | Dimension of the solution space |
>
>
>
> Do you have any other suggestions or feedback? Please let us know, and we will actively consider them.

---

> > ### Author Response · Authors · 2024-11-25
> > **Reply to Reviewer VX5p**
> >
> > Dear Reviewer VX5p,
> >
> > We sincerely appreciate the time and effort you have dedicated to reviewing our work. We understand that you have a busy schedule, and we kindly remind you that the revision deadline is approaching. In the revised version of our paper, we have **redrawn all the figures to improve their readability**. Additionally, we have provided **a notation for the algorithm section**, included in Appendix E. If you have any further questions or concerns, please do not hesitate to let us know. Moreover, if you find our response satisfactory, we would greatly appreciate your consideration of an improved rating. Thank you again for your valuable contribution!
> >
> > Best,
> >
> > The Authors

---

> > > ### Author Response · Authors · 2024-11-30
> > > **Gentle Reminder of the Rebuttal Deadline**
> > >
> > > Dear Reviewer VX5p,
> > >
> > > This is a kind reminder that the discussion phase will be ending soon. If you have any additional questions or concerns, please feel free to reach out to us. Moreover, if you are satisfied with our response, we would greatly appreciate your consideration in updating the rating. Once again, thank you for your time and valuable suggestions.
> > >
> > > Best,
> > >
> > > The Authors

---

### Official Review · Reviewer_3epy · 2024-10-31

**Soundness:** 2
**Presentation:** 3
**Contribution:** 2
**Rating:** 6
**Confidence:** 4

**Summary:**

This paper proposes a propagation technique based on directed hypergraph for data augmentation in Preferential Bayesian Optimization, which improves efficiency in the use of data and achieves competitive results to function-based BO.

**Strengths:**

(1) The paper identifies shortcomings of traditional methods in terms of under-utilisation of data and proposed a propagation method to improve data efficiency.

(2) The proposed method seem to significantly improve performance of PBO in both synthetic and real-world data.

(3) The paper is easy to follow.

**Weaknesses:**

(1) The paper claim that "existing dueling optimization methods do not make full use of the pairwise preferences collected" but lacks of support. Why augmented relation can not be learnt in original algorithms, which indicates that relation propagation is necessary? I think the current experiment(Figure 4.) is not sufficient enough.

(2) The discussion on sensitivity is cursory. The PBO performance is naturally related to the clustering performance, since the assumption that similar solutions exhibit analogous relations holds only when exact clustering results is achieved.  Why RAPBO is consistent when k varies or clustering results?

(3) No investigation on other neighbor-based augmentation methods such as linear interpolation. That is, is GMM clustering + add edges unique to this problem?

(4) The contribution focus on the proposed augmentation method.
I'm willing to raise the score if the authors can address the my first two concerns.

**Questions:**

Figure 4. (top plot) indicates that the original method have better Preference Num. than Augmented, which conflicts with the claim in the paper? Is this a typo or my understanding is wrong?

---

> ### Author Response · Authors · 2024-11-17
> **Reply to Reviewer 3epy (1/2)**
>
> We appreciate your valuable and thoughtful feedback, as well as the time you dedicated to reviewing our paper. Below, we address the questions and concerns.
>
> **Q1: Figure 4 (top plot) indicates that the original method have better Preference Num. than Augmented.**
>
> We apologize for the confusion caused by the unclear presentation in Figure 4, and we appreciate you pointing out the issue. In Figure 4 (top plot), the "number of original preferences" refers to the number of preferences that have not undergone preference propagation in each iteration, while the "number of augmented preferences" represents the **new preferences added** after the preference propagation process. Therefore, **in RAPBO, what is actually used in each iteration is the combination of original preferences and augmented preferences**. The purpose of Figure 4 is more to explore the underlying behavior of the preference propagation technique and to verify its rationality and effectiveness. We will provide a more detailed explanation of Figure 4 in the revised version to avoid any further confusion.
>
> **Q2: Why augmented relation can not be learnt in original algorithms, which indicates that relation propagation is necessary?**
>
> In previous preference based optimization works, such as PBO[1], qEUBO[2], and others, the existing pairwise preferences are directly used as datasets to train surrogate models (usually Gaussian Processes, GP). **These methods do not attempt to delve deeper into the preference dataset**, such as capturing additional potential preferential relations. This is why we say "existing dueling optimization methods do not make full use of the pairwise preferences collected".
>
> In our work, **RAPBO attempts to uncover the potential preferential relations between solutions within different preferences** through the preference propagation technique, which allows us to perform relation augmentation and make fuller use of the existing preferences throughout the preference optimization process.
>
> As for Figure 4, as mentioned in our response to Q1, it further explores the behavior of the preference propagation technique. Of course, since the "number of original preferences" refers to the number of preferences before preference propagation, we can, to some extent, regard this as the degree to which traditional methods utilize preferences. In contrast, RAPBO uses both the original preferences and augmented preferences during the optimization process, meaning that the "number of original preferences" plus the "number of augmented preferences" can be seen as the degree to which RAPBO utilizes the preferences. Therefore, **Figure 4 could verify that RAPBO makes more fuller use of the dataset compared to traditional dueling optimization methods and indicates that preference propagation is necessary**. This point has already been mentioned in the revised version of the paper (lines 430~431).

---

> > ### Author Response · Authors · 2024-11-17
> > **Reply to Reviewer 3epy (2/2)**
> >
> > **Q3: Why RAPBO is consistent when $k$ varies?**
> >
> > Thank you very much for taking the time to thoroughly read our paper and for raising such constructive question.
> >
> > To further investigate why RAPBO remains consistent as $k$ varies, we analyze the behavior of the preference propagation technique under different hyper-parameters $k$ (using the method in Figure 4).
> >
> > The table below presents the number of augmented preferences (preferences newly added after executing preference propagation) and the accuracy of the augmented preferences obtained by the preference propagation technique at different iterations, when RAPBO uses different values of $k$ on the synthetic function Griewank ($D=10$). More detailed results can be found in the revised version of the paper.
> >
> > Table 1: The mean number of augmented preferences and the accuracy of these augmented preferences of RAPBO under different hyper-parameters and iteration counts on the Griewank function.
> >
> > | k   | Metric          | 20         | 40         | 60         | 80         | 100        |
> > | :---: | :---:| :----------: | :----------: | :----------: | :----------: | :----------: |
> > | **3**   | **Augmented Preference Num.**| $26.325$| $57.500$       | $87.500$       | $117.500$      | $147.500$      |
> > |     | **Acc.** | $0.854$ | $0.865$   | $0.876$ | $0.900$ | $0.905$ |
> > | **4**   | **Augmented Preference Num.**| $22.750$      | $55.625$     | $88.500$       | $118.500$      | $146.575$    |
> > |     | **Acc.**| $0.833$ | $0.841$ | $0.901$ | $0.903$ | $0.910$ |
> > | **5**  | **Augmented Preference Num.**| $20.700$       | $57.500$       | $86.675$     | $117.650$     | $145.575$    |
> > |     |**Acc.** | $0.821$ | $0.866$ | $0.902$ | $0.929$ | $0.929$ |
> > | **6**   | **Augmented Preference Num.**| $18.550$      | $50.700$       | $81.850$      | $110.775$    | $145.575$    |
> > |     | **Acc.**| $0.837$  | $0.837$  | $0.894$  | $0.927$ | $0.932$ |
> >
> >
> > From the data in the table, we can observe that **the choice of the hyper-parameter $ k $ within a limited range does not significantly affect the number of new augmented preferences added after preference propagation, nor their accuracy**.
> >
> > Meanwhile, this property is determined by the design of the preference propagation technique. In RAPBO, as shown in Figure 1, when clustering existing solutions using GMM, we select the solutions in the cluster with the highest intra-cluster similarity as the similar solutions. Then, based on these similar solutions, we use a hypergraph to perform preference propagation. Thus, **the clustering algorithm essentially helps identify a subset of all solutions, which contains the most similar solutions**, which also explains why RAPBO maintains consistency across different values of $k$.
> >
> > The above analyses could explain why the performance of RAPBO is not sensitive to the hyper-parameter $ k $. And these points will be included in the revised version of the paper.
> >
> > **Q4: That is, is GMM clustering + add edges unique to this problem?**
> >
> > Thank you very much for your question. In RAPBO, we propose a preference propagation technique (GMM clustering + adding edges) to augment the relations within an existing preference dataset and the technique leverages the transitivity of pairwise preference (i.e., which solution is preferred). Therefore, **the preference propagation technique is specifically designed for dueling optimization**.
> >
> > The neighbor-based augmentation methods you mentioned are indeed a very effective data augmentation approach in traditional function value based methods. However, in the context of preference optimization, we can only use pairwise preferences. Directly applying neighbor-based augmentation methods could not be appropriate in this case.
> >
> > Of course, we believe that applying the ideas of neighbor-based augmentation methods to the context of preference optimization would be a very worthwhile endeavor to explore.
> >
> > [1] González, Javier, et al. "Preferential bayesian optimization." International Conference on Machine Learning. PMLR, 2017.
> >
> > [2] Astudillo, Raul, et al. "qEUBO: A decision-theoretic acquisition function for preferential Bayesian optimization." International Conference on Artificial Intelligence and Statistics. PMLR, 2023.

---

> > ### Comment · Reviewer_3epy · 2024-11-25
> >
> > Thank you for your response. I have updated my scores.

---

> > > ### Author Response · Authors · 2024-11-25
> > > **Reply to Reviewer 3epy**
> > >
> > > We sincerely thank you for your response and for raising the score. Your valuable suggestions are important in helping us improve our work, such as refining hyper-parameter analysis. We are truly grateful for your time and effort in reviewing our manuscript. Please let us know if you have any further questions.

---

### Official Review · Reviewer_4rtJ · 2024-11-04

**Soundness:** 2
**Presentation:** 2
**Contribution:** 1
**Rating:** 3
**Confidence:** 4

**Summary:**

This paper studies the problem of dueling black box optimization, an extension of black box optimization problem (or Bayesian Optimization (BO)). In the scenario of dueling BO, the agent does not have access to the function values, but can only sample two points at the same time and get only a noisy sample about whose function value is larger (i.e., a noisy comparison between these two points). This paper uses the logistic assumption for the comparison outcomes.

The authors proposed a new algorithm relation augmented preferential Bayesian optimization (RAPBO) for dueling BO, and then make numerical experiments to compare the proposed algorithm with previous algorithms. The numerical results show some superiority of the proposed algorithm than previous algorithms.

**Strengths:**

This paper studies an interesting problem and proposed a new algorithm with some novelty. The numerical results are good and give some hints about the performance of the proposed algorithm.

**Weaknesses:**

This paper has a major weakness: lack of evidence to show the improvements of the proposed algorithm than previous algorithms. This paper does not have any theoretical conclusions of the proposed algorithm. It does have numerical experiments, but from the writing of the paper and small scales of the experiments, these experiments do not seem to be a standard benchmark for the dueling BO area. Hence, from the writing of the paper, it does not have enough evidence to demonstrate the significance of this paper's results.

**Questions:**

Is there any theoretical conclusion about the proposed algorithm, and how does it compare to existing algorithms?

---

> ### Author Response · Authors · 2024-11-17
> **Reply to Reviewer 4rtJ (1/2)**
>
> Thank you for your comprehensive review and insightful feedback. We have carefully considered your feedback and provided detailed responses below.
>
> **Q1: Lack of evidence to show the improvements of the proposed algorithm than previous algorithms.**
>
> In the paper, we design a series of experiments to verify that RAPBO achieves superior performance compared to previous algorithms. First, we showcase the effectiveness and superiority of RAPBO through experiments on a range of synthetic functions and real-world datasets (**see Figures 2 and 3**). As shown in the results, **RAPBO shows significant improvements over the previous methods in all cases**. For a more detailed results, please see the Table 1 and Table 2.
>
>
> Table 1: The detailed results of dueling optimization methods on synthetic functions. In each column, an entry with the best mean value is marked in bold and underline for the runner-up. If the mean value of the best method significantly differs from the runner-up, passing a t-test with a significance level of $0.05$, then we denote it with "*" at the corresponding position.
>
> | **Method**  | **Rosenbrock**                                    | **Dixon**                                        | **Griewank**                                    | **Levy**                                          | **Schwefel**                                      | **Sphere**                                        |
> |:-----------:|:------------------------------------------------:|:------------------------------------------------:|:------------------------------------------------:|:------------------------------------------------:|:------------------------------------------------:|:------------------------------------------------:|
> | PBO         | $-66175.670 \pm 31607.440$                       | $-34068.125 \pm 18046.893$                       | $\underline{-1.578 \pm 0.180}$                        | $-27.021 \pm 6.795$                              | $\underline{-4085.577 \pm 25.830}$                    | $-25.920 \pm 6.396$                              |
> | KSS         | $\underline{-43576.960 \pm 30242.922}$                | $\underline{-32522.400 \pm 17915.740}$                | $-1.624 \pm 0.168$                               | $-23.338 \pm 7.265$                              | $-4088.018 \pm 19.577$                           | $\underline{-21.097 \pm 6.080}$                       |
> | qEUBO       | $-71249.730 \pm 45543.810$                       | $-33604.113 \pm 18374.818$                       | $-1.824 \pm 0.147$                               | $\underline{-21.008 \pm 8.093}$                       | $-4094.577 \pm 41.635$                           | $-24.646 \pm 10.413$                             |
> | COMP-UCB    | $-85650.450 \pm 51355.145$                       | $-44837.203 \pm 16389.234$                       | $-1.748 \pm 0.168$                               | $-26.435 \pm 6.614$                              | $-4100.247 \pm 20.779$                           | $-30.497 \pm 7.499$                              |
> | RAPBO       | $\boldsymbol{-18416.717 \pm  11828.153^*}$                      | $\boldsymbol{-15456.545 \pm 10959.241^*}$                 | $\boldsymbol{-1.3798 \pm 0.100^*}$                         | $\boldsymbol{-18.388 \pm 3.880}$                           | $\boldsymbol{-4072.887 \pm 15.641} $                       | $\boldsymbol{-16.432 \pm 3.480^*} $                        |
>
>
> Table2: The detailed results of dueling optimization methods on real-world datasets. In each column, an entry with the best mean value is marked in bold and underline for the runner-up. If the mean value of the best method significantly differs from the runner-up, passing a t-test with a significance level of $0.05$, then we denote it with "*" at the corresponding position.
>
> | **Method**  | **RobotPush**                           | **Cassini1-MINLP**                        | **Sagas**                               |
> |:-----------:|:--------------------------------------:|:----------------------------------------:|:--------------------------------------:|
> | PBO         | $3.881 \pm 1.221$                      | $\underline{-58.359 \pm 11.105}$               | $\underline{-2.117 \pm 0.730}$               |
> | KSS         | $3.909 \pm 1.000$                      | $-70.781 \pm 25.648$                      | $-3.382 \pm 0.677$                     |
> | qEUBO       | $3.204 \pm 1.038$                      | $-83.197 \pm 25.342$                      | $-3.922 \pm 1.071$                     |
> | COMP-UCB    | $\underline{3.918 \pm 1.247}$               | $-81.956 \pm 27.205$                      | $-3.721 \pm 0.825$                     |
> | RAPBO       | $\boldsymbol{4.302 \pm 1.205}$                 | $\boldsymbol{-46.850 \pm 16.175^*}$             | $\boldsymbol{-1.478 \pm 0.169^*}$              |

---

> > ### Author Response · Authors · 2024-11-17
> > **Reply to Reviewer 4rtJ (2/2)**
> >
> > **Q2: From the writing of the paper and small scales of the experiments, these experiments do not seem to be a standard benchmark for the dueling BO area.**
> >
> > Thank you for your thoughtful feedback.
> >
> > One area related to dueling optimization is dueling bandits, which is typically studied in the context of K-armed bandits, whereas dueling optimization is performed on a finite K-arms in continuous solution spaces. In dueling bandits, the work most closely related to dueling optimization is KSS [1], but it mainly conducts experiments on synthetic functions.
> >
> > Additionally, PBO [2] is a classic work in preferential Bayesian optimization, but it only conducts experiments on synthetic functions. Subsequent methods, such as skewGP-PBO [3] and HB [4], also primarily conduct experiments on synthetic functions, and the synthetic functions used between these methods differ.
> >
> > Therefore, to the best of our knowledge, **previous work on dueling optimization primarily conduct experiments on synthetic functions**. In this context, to thoroughly evaluate the performance of our algorithm, we conduct experiments not only on synthetic functions from previous work, such as Levy and Rosenbrock, but also on additional synthetic functions from different optimization problem types.
> >
> > Regarding real-world datasets, **we select commonly used real-world datasets that are both suitable for solving using dueling optimization and aligned with our motivation**, such as noisy motion control task and spacecraft trajectory optimization problems, which are costly and time-consuming to observe. In contrast, previous works like KSS use real-world datasets that do not align with our motivation.
> >
> > **Q3: This paper does not have any theoretical conclusions of the proposed algorithm.**
> >
> > The main contribution of this paper lies in filling the optimization performance gap between preference based and function value based methods, for which we propose RAPBO. In our research, we initially attempted to derive theoretical results for our algorithm within the dueling bandit framework. However, in the context of dueling optimization, **the objective function does not always satisfy common assumptions such as linearity or convexity**, which prevents us from directly performing effective theoretical derivations within the existing framework. As a result, we had to approach the problem from alternative angles and explore different perspectives for theoretical analysis.
> >
> > Furthermore, the preference propagation technique we proposed integrates **GMM clustering and propagates preferences through a hypergraph structure, which further complicates the theoretical derivation**. Despite numerous attempts, unfortunately, we have been unable to derive valuable theoretical insights regarding the preference propagation method. Nevertheless, in future work, we will continue to focus on this direction, exploring potential theoretical frameworks with the aim of obtaining deeper theoretical conclusions.
> >
> > [1] Sui, Yanan, et al. "Multi-dueling bandits with dependent arms." arXiv preprint arXiv:1705.00253 (2017).
> >
> > [2] González, Javier, et al. "Preferential bayesian optimization." International Conference on Machine Learning. PMLR, 2017.
> >
> > [3] Benavoli, Alessio, Dario Azzimonti, and Dario Piga. "Preferential bayesian optimisation with skew gaussian processes." Proceedings of the Genetic and Evolutionary Computation Conference Companion. 2021.
> >
> > [4] Takeno, Shion, Masahiro Nomura, and Masayuki Karasuyama. "Towards practical preferential Bayesian optimization with skew Gaussian processes." International Conference on Machine Learning. PMLR, 2023.

---

> > > ### Author Response · Authors · 2024-11-25
> > > **Reply to Reviewer 4rtJ**
> > >
> > > Dear Reviewer 4rtJ,
> > >
> > > We sincerely appreciate the time and effort you have dedicated to reviewing our work. We understand that you have a busy schedule, and we kindly remind you that the revision deadline is approaching. During the rebuttal period, we diligently addressed your concerns by providing point-to-point responses, which encompassed **detailed explanations of the experimental environment (synthetic functions + real-world tasks) and setup**, along with an **in-depth statistical analysis of the experimental results using t-tests**. If you have any further questions or concerns, please do not hesitate to let us know. Moreover, if you find our response satisfactory, we would greatly appreciate your consideration of an improved rating. Thank you again for your valuable contribution!
> > >
> > > Best,
> > >
> > > The Authors

---

> > > > ### Author Response · Authors · 2024-11-30
> > > > **Gentle Reminder of the Rebuttal Deadline**
> > > >
> > > > Dear Reviewer 4rtJ,
> > > >
> > > > This is a kind reminder that the discussion phase will be ending soon. If you have any additional questions or concerns, please feel free to reach out to us. Moreover, if you are satisfied with our response, we would greatly appreciate your consideration in updating the rating. Once again, thank you for your time and valuable suggestions.
> > > >
> > > > Best,
> > > >
> > > > The Authors

---

> > > > > ### Comment · Reviewer_4rtJ · 2024-12-02
> > > > > **Response to the authors**
> > > > >
> > > > > In my opinion, the results are not confident enough to claim that the new algorithms beat the previous ones. We need more confident evidence. Thanks

---

> ### Author Response · Authors · 2024-12-02
> **Response to Reviewer 4rtJ**
>
> Thank you for your valuable time and comments.

---

### Official Review · Reviewer_zwz1 · 2024-11-04

**Soundness:** 2
**Presentation:** 3
**Contribution:** 2
**Rating:** 5
**Confidence:** 4

**Summary:**

This work studied the dueling optimization problems under the framework of preferential Bayesian optimization (PBO). The key finding was that by leveraging directed hypergraphs to augment the relations among preference dataset, the optimization efficiency of (PBO) could be improved, and even competitive with the real-value-based Bayesian optimization method. Both synthetic functions and real-world tasks were considered to validate the effectiveness.

**Strengths:**

1. The idea of utilizing a directed hypergraph to model and augment the potential relations between solution seems noval.
2. The paper is easy-to-follow. It is easy to clarify the motivations and contributions.
3. The sample efficiency of dueling optimization problems is important in recommender and human-machine interaction systems.

**Weaknesses:**

1. The motivations may not be convincing. The dueling optimization problems were originally addressed by dueling bandits algorithms. In PBO, which considered continuous search space and the Bradley-Terry-Luce model, the Gaussian processes (GPs) was introduced with kernels and priors that had already defined the relation among solution in the latent space. Therefore, in my opinion, what this work did is to supplement the characterization of correlation that has been quantified by priors and kernel functions. If so, I think a more elegent way is  to modify the format of kernel functions and strictly follow the Bayes rule, rather than performing preference propagation following by constructing two GPs in a single round, as presented from Line 167 to Line 170 in Algorithm 1, which not only increases computational complexity but also weakens the principle of the Bayesian optimization.
3. The contributions, a.k.a. the introduction of GMM and directed hypergraph, were somewhat noval but not strong enough. No theoretical analysis nor design principles were used to underpin the rationality of the proposed algorithm.
2. The experiment results are not satisfactory. The big variance implies that the proposed method may not statistically outperform other baselines in most cases. Methods such as Wilcoxon rank sum test could be used to demonstrate this point.

**Questions:**

1. What is the essential difference between introducing the directed hypergraph and calibrating the kernel and prior functions of GPs?
2. Is there any evidence that value-based Bayesian optimization should be more efficient than pair-wise comparison PBO?
3. Compared with GP-UCB or PBO, what is the the computational complexity of RAPBO?

---

> ### Author Response · Authors · 2024-11-17
> **Reply to Reviewer zwz1 (1/3)**
>
> We thank you for the useful and insightful feedback and for taking the time to review our paper. We address questions and concerns below.
>
> **Q1: The GPs with kernels and priors have already defined the relation among solutions in the latent space.**
>
> Thank you for your insightful feedback, which has prompted us to think more deeply about the relationship and distinctions between our preference propagation technique and GPs with kernels.
>
> Unlike GP with kernels, RAPBO tries to capture the potential relations between different pairwise preferences, which uses a preference propagation technique to propagate preferences over an existing dataset, thereby augmenting the relations.
>
> Therefore, **the role of our preference propagation technique is to perform relation augmentation on the dataset through preference propagation**. In other words, the preference propagation technique provides a more comprehensive dataset for the surrogate model (GPs with kernels), which does not weaken the principle of Bayesian optimization.
>
> **Q2: The contributions are not strong enough. & No theoretical analysis nor design principles are used to underpin the rationality of the proposed algorithm.**
>
> Thank you for your thoughtful feedback.
>
> Although dueling optimization relies solely on pairwise preferences without using function values, an obvious optimization performance gap exists between preference based and function value based methods. Therefore, we propose RAPBO, which aims to **fill the optimization performance gap between preference based and function value based methods**. To achieve this, we start from the perspective of **how to make fuller use of pairwise preferences** and introduce the preference propagation technique to uncover the potential preferential relations between solutions within different preferences in the dueling optimization process.
>
> Meanwhile, we have conducted extensive experiments to verify the effectiveness and rationality of the proposed algorithm.
>
> 1. **Effectiveness and superiority of RAPBO.** We conduct experiments on a series of synthetic functions (**see Figure 2**), which collectively cover various types of optimization problems. In addition, to further highlight the effectiveness and rationality of RAPBO, we also test it on real-world tasks such as motion control and spacecraft trajectory optimization (**see Figure 3**).
>
> 2. **Utilization of RAPBO.** What's more, we delve deeper into the preference propagation technique (**see Figure 4**). The experimental results show that the preference propagation technique not only gathers additional preferences from existing datasets but also ensures that the newly added preferences maintain a high level of accuracy, further confirming the rationality of RAPBO.
>
> 3. **Compare with function value based methods.** In line with our motivation, we compare preference based methods with function value based methods (**see Figure 5**). The experimental results verify that, under the same cost budget, RAPBO is competitive with, or even surpasses, function value based Bayesian optimization methods in terms of optimization performance.
>
> As for the theoretical analysis, we have made every effort to conduct it. Dueling optimization is closely related to the field of dueling bandits, and we initially sought to leverage the theoretical framework of dueling bandits to derive theoretical results. However, **in the context of dueling optimization, the objective function often fails to meet common assumptions** such as linearity or convexity, which forces us to explore theoretical analysis from alternative perspectives. Additionally, the proposed preference propagation technique, which incorporates GMM clustering and propagates preferences through a hypergraph structure, **introduces additional complexities that further challenge theoretical analysis**. Despite extensive efforts, we have not yet arrived at meaningful theoretical insights. Nevertheless, we remain committed to pursuing theoretical advancements in future work.

---

> > ### Author Response · Authors · 2024-11-17
> > **Reply to Reviewer zwz1 (2/3)**
> >
> > **Q3: The big variance implies that the proposed method may not statistically outperform other baselines in most cases.**
> >
> > In our experiments, RAPBO and other preference based methods all exhibit relatively high standard deviations in experiments on both synthetic functions and real-world datasets. This is because we do not standardize the results. However, **RAPBO exhibits the lowest standard deviation in most experiments** and consistently provides the best performance across all experiments.
> >
> > To verify that RAPBO statistically outperforms other baselines in most cases, we perform t-tests with a significance level of $0.05$.
> >
> > Table 1: The detailed results of dueling optimization methods on synthetic functions. In each column, an entry with the best mean value is marked in bold and underline for the runner-up. If the mean value of the best method significantly differs from the runner-up, passing a t-test with a significance level of $0.05$, then we denote it with "*" at the corresponding position.
> >
> > | **Method**  | **Rosenbrock**                                    | **Dixon**                                        | **Griewank**                                    | **Levy**                                          | **Schwefel**                                      | **Sphere**                                        |
> > |:-----------:|:------------------------------------------------:|:------------------------------------------------:|:------------------------------------------------:|:------------------------------------------------:|:------------------------------------------------:|:------------------------------------------------:|
> > | PBO         | $-66175.670 \pm 31607.440$                       | $-34068.125 \pm 18046.893$                       | $\underline{-1.578 \pm 0.180}$                        | $-27.021 \pm 6.795$                              | $\underline{-4085.577 \pm 25.830}$                    | $-25.920 \pm 6.396$                              |
> > | KSS         | $\underline{-43576.960 \pm 30242.922}$                | $\underline{-32522.400 \pm 17915.740}$                | $-1.624 \pm 0.168$                               | $-23.338 \pm 7.265$                              | $-4088.018 \pm 19.577$                           | $\underline{-21.097 \pm 6.080}$                       |
> > | qEUBO       | $-71249.730 \pm 45543.810$                       | $-33604.113 \pm 18374.818$                       | $-1.824 \pm 0.147$                               | $\underline{-21.008 \pm 8.093}$                       | $-4094.577 \pm 41.635$                           | $-24.646 \pm 10.413$                             |
> > | COMP-UCB    | $-85650.450 \pm 51355.145$                       | $-44837.203 \pm 16389.234$                       | $-1.748 \pm 0.168$                               | $-26.435 \pm 6.614$                              | $-4100.247 \pm 20.779$                           | $-30.497 \pm 7.499$                              |
> > | RAPBO       | $\boldsymbol{-18416.717 \pm  11828.153^*}$                      | $\boldsymbol{-15456.545 \pm 10959.241^*}$                 | $\boldsymbol{-1.3798 \pm 0.100^*}$                         | $\boldsymbol{-18.388 \pm 3.880}$                           | $\boldsymbol{-4072.887 \pm 15.641} $                       | $\boldsymbol{-16.432 \pm 3.480^*} $                        |
> >
> >
> > Table2: The detailed results of dueling optimization methods on real-world datasets. In each column, an entry with the best mean value is marked in bold and underline for the runner-up. If the mean value of the best method significantly differs from the runner-up, passing a t-test with a significance level of $0.05$, then we denote it with "*" at the corresponding position.
> >
> > | **Method**  | **RobotPush**                           | **Cassini1-MINLP**                        | **Sagas**                               |
> > |:-----------:|:--------------------------------------:|:----------------------------------------:|:--------------------------------------:|
> > | PBO         | $3.881 \pm 1.221$                      | $\underline{-58.359 \pm 11.105}$               | $\underline{-2.117 \pm 0.730}$               |
> > | KSS         | $3.909 \pm 1.000$                      | $-70.781 \pm 25.648$                      | $-3.382 \pm 0.677$                     |
> > | qEUBO       | $3.204 \pm 1.038$                      | $-83.197 \pm 25.342$                      | $-3.922 \pm 1.071$                     |
> > | COMP-UCB    | $\underline{3.918 \pm 1.247}$               | $-81.956 \pm 27.205$                      | $-3.721 \pm 0.825$                     |
> > | RAPBO       | $\boldsymbol{4.302 \pm 1.205}$                 | $\boldsymbol{-46.850 \pm 16.175^*}$             | $\boldsymbol{-1.478 \pm 0.169^*}$              |
> >
> >
> > As shown in the table, **RAPBO statistically outperforms other baselines in most cases**. These results have already been added in the revised version, as detailed in Appendix D.

---

> > > ### Author Response · Authors · 2024-11-17
> > > **Reply to Reviewer zwz1 (3/3)**
> > >
> > > **Q4: Is there any evidence that value based Bayesian optimization should be more efficient than pairwise comparison PBO?**
> > >
> > > Thank you for your thoughtful question.
> > >
> > > Function value based methods require continuous observations of the objective function during the optimization process, which may not be feasible in certain cases (e.g., when the evaluating function values is expensive or even unavailable). In this context, preference based methods are proposed, where only pairwise preferences are needed during the optimization process. To the best of our knowledge, there has been little work in the literature that compares both function value based methods and preference based methods together. However, considering that function value based methods, such as GP-UCB, can use the function values, **which are regarded as strong baselines**, while preference based methods can only use pairwise preferences (i.e., which solution is preferred), we believe that **there exists an optimization performance gap between the two categories of methods, and this gap is intuitive**.
> > >
> > > At the same time, from Figure 5, we can observe that under a fixed budget, regardless of whether we set the cost of observing function values to be twice or 1.5 times that of comparing a pair of duels, the performance of the function value based method (GP-UCB) ultimately surpasses that of PBO. **The experimental results also verify the existence of an optimization performance gap between function value based and preference based methods.**
> > >
> > > **Q5: Compared with GP-UCB or PBO, what is the the computational complexity of RAPBO?**
> > >
> > > First, we compare the computational complexity of RAPBO and PBO. RAPBO's computational complexity is more complex, primarily due to the preference propagation technique. In this technique, we need to maintain an additional surrogate model and achieve relation augmentation through GMM clustering and hypergraphs. The introduction of hypergraphs further reduces the computational complexity of the preference propagation technique (as shown in Section 4.3). However, **since our budget in dueling optimization is limited, the preference propagation technique does not introduce significant computational complexity**.
> > >
> > > Next, we compare the computational complexity of RAPBO and GP-UCB. Clearly, **GP-UCB has a higher computational complexity than RAPBO**, primarily due to differences in the evaluation phase. GP-UCB is a function value based optimization method, and during the optimization process, GP-UCB needs to continuously observe the objective function, which is very expensive and time-consuming. In contrast, RAPBO only requires pairwise preferences during the optimization process, which results in lower computational complexity for RAPBO.

---

> > > ### Comment · Reviewer_4rtJ · 2024-11-20
> > > **Response to the rebuttal**
> > >
> > > Thanks for your response.
> > > I think for a paper to be accepted by a top conference like ICLR, we need to be conservative to claim that a proposed algorithm is better than existing ones. If no theoretical proofs to show the superiority, it is necessary to have very confident numerical evidence.
> > > For the experiments demonstrated in this paper, a series of questions are unclear.
> > > 1. Are the real-world data sets commonly accepted? Like in other areas, people have widely accepted datasets like imagenet to compare with. This point is not well stated in this paper.
> > > 2. Are the experiments of large enough scales, in both number of trials and scales of the datasets. From the paper, the experiments only have several to 100 trials, and the number of iterations is also limited
> > > 3. Are these experiment results confident enough? In the real-world data based experiments, the deviations are too large to confidently claim whether the new algorithms are truly better than existing ones.

---

> > ### Comment · Reviewer_zwz1 · 2024-11-22
> >
> > Thanks for the authors' feedback. My concerns remain in the following aspects.
> > 1. As a heuristic design without theoretical insights, the proposed algorithm, namely the preference propagation, is expected to significantly outperform other baselines in most cases. However, the number of benchmarks are limited (6 synthetic problems and 3 toy problems in ML community), and the trajectories are close especially in the early 50 evaluations. The results are somewhat weird because if data-augmentation works, the algorithm should be better when data is limited. In all, stating that the proposed algorithm shows "superiority" is incorrect.
> > 2. Compare with function value based methods, there is no thumb rule to determine the budget correlation between pairwise comparison and value observation as far as I know. The experimental results are less meaningful to the practationers.

---

> ### Author Response · Authors · 2024-11-20
> **Reply to the Reviewer 4rtJ (1/3)**
>
> Thank you for your valuable questions. We fully understand your concerns and we will address your doubts regarding the experiments.
>
> **Q1: Are the real-world datasets commonly accepted?**
>
> Thank you for your insightful question.
>
> Regarding the experimental section, we believe that **our choice of synthetic functions and real-world datasets is reasonable and commonly accepted**.
>
> First, **our real-world datasets are commonly accepted**. Specifically, the noisy motion control task RobotPush, widely used in Bayesian optimization **like EBO [1] and TurBO [2]**, is a well-established real-world dataset. Additionally, the remaining two datasets, which involve spacecraft trajectory optimization problems, are also widely applied in the optimization domain. Meanwhile, as stated in our paper (lines 394–399), these datasets are all suitable for dueling optimization.
>
> Then, as mentioned in our previous response, prior works such as KSS [3] and PBO [4] on dueling optimization primarily conduct experiments on synthetic functions. Accordingly, we initially use synthetic functions from these studies and also conduct experiments on additional synthetic functions from different types of optimization problems.
>
> Therefore, **these real-world datasets are both commonly accepted and highly suitable for solving using dueling optimization, while also aligning with our motivation**.
>
> **Q2: Are the experiments of large enough scales, in both number of trials and scales of the datasets?**
>
> Thank you for your thoughtful question.
>
> In Bayesian optimization, we assume that observing the objective function is both **time-consuming and expensive, and we can only perform optimization within a limited budget** [5] [6]. Consequently, in our experiments, we also set a finite budget for evaluation, consistent with previous dueling Bayesian optimization works such as HB [7] and qEUBO [8].
>
> Moreover, whether it is our method or the compared methods, their performance curves typically rise rapidly during the early stages of optimization and **gradually converge later, showing no further improvement**. This pattern indicates that the budget of 100 we allocated is both sufficient and reasonable.
>
>  **Q3: Are these experiment results confident enough?**
>
> Thank you for your thoughtful question.
>
> In our experiments, RAPBO and other preference based methods exhibit relatively high standard deviations. **This is because the scales of the data vary across different tasks**. However, RAPBO consistently provides the best performance across all experiments, and in most cases, **the standard deviation of RAPBO is the smallest** compared to other methods such as PBO and KSS.
>
> To verify that RAPBO truly outperforms other baselines in most cases, **we have performed t-tests with a significance level of $0.05$**. As shown in Tables 1 and 2, we denote the best method with '*' at the corresponding position if its mean value significantly differs from that of the runner-up. The results show that **RAPBO statistically outperforms other baselines in the most of cases**.

---

> ### Author Response · Authors · 2024-11-20
> **Reply to the Reviewer 4rtJ (2/3)**
>
> Table 1: The detailed results of dueling optimization methods on synthetic functions. In each column, an entry with the best mean value is marked in bold and underline for the runner-up. If the mean value of the best method significantly differs from the runner-up, passing a t-test with a significance level of $0.05$, then we denote it with "*" at the corresponding position.
>
> | **Method**  | **Rosenbrock**                                    | **Dixon**                                        | **Griewank**                                    | **Levy**                                          | **Schwefel**                                      | **Sphere**                                        |
> |:-----------:|:------------------------------------------------:|:------------------------------------------------:|:------------------------------------------------:|:------------------------------------------------:|:------------------------------------------------:|:------------------------------------------------:|
> | PBO         | $-66175.670 \pm 31607.440$                       | $-34068.125 \pm 18046.893$                       | $\underline{-1.578 \pm 0.180}$                        | $-27.021 \pm 6.795$                              | $\underline{-4085.577 \pm 25.830}$                    | $-25.920 \pm 6.396$                              |
> | KSS         | $\underline{-43576.960 \pm 30242.922}$                | $\underline{-32522.400 \pm 17915.740}$                | $-1.624 \pm 0.168$                               | $-23.338 \pm 7.265$                              | $-4088.018 \pm 19.577$                           | $\underline{-21.097 \pm 6.080}$                       |
> | qEUBO       | $-71249.730 \pm 45543.810$                       | $-33604.113 \pm 18374.818$                       | $-1.824 \pm 0.147$                               | $\underline{-21.008 \pm 8.093}$                       | $-4094.577 \pm 41.635$                           | $-24.646 \pm 10.413$                             |
> | COMP-UCB    | $-85650.450 \pm 51355.145$                       | $-44837.203 \pm 16389.234$                       | $-1.748 \pm 0.168$                               | $-26.435 \pm 6.614$                              | $-4100.247 \pm 20.779$                           | $-30.497 \pm 7.499$                              |
> | RAPBO       | $\boldsymbol{-18416.717 \pm  11828.153^*}$                      | $\boldsymbol{-15456.545 \pm 10959.241^*}$                 | $\boldsymbol{-1.3798 \pm 0.100^*}$                         | $\boldsymbol{-18.388 \pm 3.880}$                           | $\boldsymbol{-4072.887 \pm 15.641} $                       | $\boldsymbol{-16.432 \pm 3.480^*} $                        |
>
>
> Table2: The detailed results of dueling optimization methods on real-world datasets. In each column, an entry with the best mean value is marked in bold and underline for the runner-up. If the mean value of the best method significantly differs from the runner-up, passing a t-test with a significance level of $0.05$, then we denote it with "*" at the corresponding position.
>
> | **Method**  | **RobotPush**                           | **Cassini1-MINLP**                        | **Sagas**                               |
> |:-----------:|:--------------------------------------:|:----------------------------------------:|:--------------------------------------:|
> | PBO         | $3.881 \pm 1.221$                      | $\underline{-58.359 \pm 11.105}$               | $\underline{-2.117 \pm 0.730}$               |
> | KSS         | $3.909 \pm 1.000$                      | $-70.781 \pm 25.648$                      | $-3.382 \pm 0.677$                     |
> | qEUBO       | $3.204 \pm 1.038$                      | $-83.197 \pm 25.342$                      | $-3.922 \pm 1.071$                     |
> | COMP-UCB    | $\underline{3.918 \pm 1.247}$               | $-81.956 \pm 27.205$                      | $-3.721 \pm 0.825$                     |
> | RAPBO       | $\boldsymbol{4.302 \pm 1.205}$                 | $\boldsymbol{-46.850 \pm 16.175^*}$             | $\boldsymbol{-1.478 \pm 0.169^*}$              |
>
> We hope these responses can resolve your issues and dispel your doubts. Thank you.

---

> > ### Author Response · Authors · 2024-11-20
> > **Reply to the Reviewer 4rtJ (3/3)**
> >
> > [1] Wang, Zi, et al. "Batched large-scale Bayesian optimization in high-dimensional spaces." International Conference on Artificial Intelligence and Statistics. PMLR, 2018.
> >
> > [2] Eriksson, David, et al. "Scalable global optimization via local Bayesian optimization." Advances in neural information processing systems 32 (2019).
> >
> > [3] Sui, Yanan, et al. "Multi-dueling bandits with dependent arms." arXiv preprint arXiv:1705.00253 (2017).
> >
> > [4] González, Javier, et al. "Preferential bayesian optimization." International Conference on Machine Learning. PMLR, 2017.
> >
> > [5] Frazier, Peter I. "A tutorial on Bayesian optimization." arXiv preprint arXiv:1807.02811 (2018).
> >
> > [6] Frazier, Peter I. "Bayesian optimization." Recent advances in optimization and modeling of contemporary problems. Informs, 2018. 255-278.
> >
> > [7] Takeno, Shion, Masahiro Nomura, and Masayuki Karasuyama. "Towards practical preferential Bayesian optimization with skew Gaussian processes." International Conference on Machine Learning. PMLR, 2023.
> >
> > [8] Astudillo, Raul, et al. "qEUBO: A decision-theoretic acquisition function for preferential Bayesian optimization." International Conference on Artificial Intelligence and Statistics. PMLR, 2023.

---

> ### Author Response · Authors · 2024-11-24
> **Reply to the Reviewer zwz1 (1/2)**
>
> Thank you for your valuable feedback. We understand your concerns and would like to clarify them.
>
> **Q1: The number of benchmarks are limited, and the trajectories are close especially in the early 50 evaluations.**
>
> Thank you for your thoughtful feedback.
>
> Regarding the experiments, we use **more synthetic functions and real-world datasets** compared to previous works such PBO [1], KSS [2] and skewGP-PBO [3] and we believe that **the experiments on synthetic functions and real-world datasets are sufficient to verify the performance of our algorithm**.
>
> For the synthetic functions, we select **a reasonable and sufficient set of synthetic functions** to evaluate the performance of our algorithm. First, we conduct experiments on synthetic functions such as Levy and Rosenbrock, which are **widely used** in prior work like PBO [1] and skewGP-PBO [3]. Additionally, the synthetic functions **cover a diverse range of characteristics**, including multimodal landscapes (Dixon, Schwefel), complex terrains (Levy), periodic variations (Griewank), and convex optimization (Sphere, Rosenbrock). Therefore, these synthetic functions are sufficient and representative, allowing us to verify the algorithm's performance across  various scenarios.
>
> Although previous works such as [1] and [3] primarily focus on synthetic functions, we also **conduct comprehensive real-world scenarios** to show the performance of RAPBO, such as RobotPush, Cassini1-MINLP, and Sagas. These real-world tasks are **both suitable for dueling optimization and aligned with our research motivation** as mentioned in lines 394-399. For example, a key challenge in real-world interplanetary space mission trajectory design tasks (such as Cassini1-MINLP and Sagas) is the difficulty of directly evaluate the quality of trajectories. However, comparing different trajectories is inexpensive, accurate, and more feasible. On the other hand, **these representative tasks show the complexity and diversity of the real-world scenarios**, and the deployment of real-world experiments can better analysis the comprehensive and robust performance of our algorithm.
>
> In the comparative experiments, **RAPBO exhibits a performance trend that aligns with our expectations**. During the early stages of optimization, its performance curve is similar to that of other methods. However, as other methods gradually converge, **RAPBO continues to improve steadily and achieves the best results**. As shown in Figure 4, this behavior is consistent with our analysis. In the early stages of optimization, the number of pairwise preferences available in the dataset is quite limited. As a result, our preference propagation technique cannot extract many preferential relations from the dataset, leading to similar performance between RAPBO and other methods. However, when the number of iterations reaches around 50, a sufficient number of pairwise preferences have been accumulated in the dataset. **At this point, our preference propagation technique can play a more significant role, extracting more potential preferential relations from the dataset, thereby achieving relational augmentation of the dataset.** This is the primary reason why RAPBO shows stable improvements during the later stages of optimization.

---

> ### Author Response · Authors · 2024-11-24
> **Reply to the Reviewer zwz1 (2/2)**
>
> **Q2: There is no thumb rule to determine the budget correlation between pairwise comparison and value observation.**
>
> Thank you for your feedback.
>
> Regarding how to determine the budget correlation between pairwise comparison and value observation, it is clear that evaluating the function value is more expensive than comparing a pair of solutions. Therefore, **we follow the experimental setup of PE-DPO** [4], where the cost of evaluating the function value is set to be **twice** that of comparing a pair of solutions. To further verify that the performance of RAPBO under a fixed budget can match that of value based methods, we conduct an additional experiment where the cost of evaluating the function value is set to be **1.5 times** that of comparing a pair of solutions.
>
> **Since the ratios we set are relatively small, and in real-world scenarios, evaluating the function value is often much more costly than comparing a pair of solutions**, our conclusion — "under the same cost budget, RAPBO is competitive with or even surpasses function value-based Bayesian optimization methods with respect to optimization performance" — is **reasonable and meaningful to practitioners**.
>
>
> [1] González, Javier, et al. "Preferential bayesian optimization." International Conference on Machine Learning. PMLR, 2017.
>
> [2] Sui, Yanan, et al. "Multi-dueling bandits with dependent arms." arXiv preprint arXiv:1705.00253 (2017).
>
>
> [3] Benavoli, Alessio, Dario Azzimonti, and Dario Piga. "Preferential bayesian optimisation with skew gaussian processes." Proceedings of the Genetic and Evolutionary Computation Conference Companion. 2021.
>
> [4] "High-dimensional dueling optimization with preference embedding." Proceedings of the AAAI Conference on Artificial Intelligence. Vol. 37. No. 9. 2023.

---

> > ### Author Response · Authors · 2024-11-30
> > **Gentle Reminder of the Rebuttal Deadline**
> >
> > Dear Reviewer zwz1,
> >
> > This is a kind reminder that the discussion phase will be ending soon. If you have any additional questions or concerns, please feel free to reach out to us. Moreover, if you are satisfied with our response, we would greatly appreciate your consideration in updating the rating. Once again, thank you for your time and valuable suggestions.
> >
> > Best,
> >
> > The Authors

---

> > > ### Comment · Reviewer_zwz1 · 2024-12-01
> > > **Response to the Authors**
> > >
> > > Thank you for your response. I appreciate the novalty of the propagation techniques, which I believe deserves a more indepth analysis regarding its performance and data augmentation behavior. In particular, the visualization of wihch data were generated, and how the proposed method can be robust to misleading gerated data is under-explored.
> > >
> > > Other contributions could be over-claimed such as the comparison to value-based evaluation. In recent machine learning research, the dueling optimization is used to fine-tune the large-language model, such as RLHF (it is not a matter whether this is conducted online or offline). Dueling optimization has also been widely used for eliciting human preference in recommender systems and human-computer interaction. These are benchmarks particularly useful for practationers. In these case, the cognitive load for users to compare solutions or give solutions a value is not measurable and quantifiable. Either value-based or pairwise-based problems are considered independently in most real applications.
> > >
> > > My concern to this work is the lack of indepth analysis for the propagation techniques. I think the paper should be reorganized to highlight this point and present more evidence (not only the final performance, but also the specific behavior) to support its effectiveness.
> > >
> > > I will respectfully keep my score and once again appreciate the authors' feedback.

---

> ### Author Response · Authors · 2024-12-01
> **Response to Reviewer zwz1**
>
> Thank you for your detailed review and valuable feedback on our work. We greatly appreciate your comments, particularly regarding the in-depth analysis of the propagation technique. We will carefully consider your suggestions to further improve the content of the paper. While we understand your concerns regarding some of the contributions, we are still grateful for your insights, which will help us enhance the quality of our work and lay a solid foundation for this research. Once again, thank you for your time and feedback.
>
> Best regards

---

### Author Response · Authors · 2024-11-17
**General Response to Reviewers and Revision Submitted.**

We would like to express our sincere gratitude to all the reviewers for their valuable feedback and constructive suggestions. We are encouraged that they find our proposed methods for relation augmentation to be novel (**Reviewers zwz1**, **Reviewers 4rtj**), numerous experiments are designed (**Reviewers VX5p**, **Reviewers 3epy**), and some hints for the algorithm are provided (**Reviewers 4rtj**). In response to the reviewers' comments, we have made revisions to the manuscript. Below, we provide a summary of the key revisions (highlighted in blue text in the PDF). Additionally, we have made some minor formatting adjustments to comply with the page limit. Detailed responses to each reviewer’s comments are provided separately.

**The key revisions are as follows:**
1. We have conducted additional experiments on the hyper-parameters to further analyze the reasons behind RAPBO's stable performance under different values of $k$. Detailed modifications can be found in Appendix B.

2. We have conducted additional analysis on the results of the utilization part in the experiment (lines 430~431).

3. In the Appendix D, We add more detailed results of RAPBO and other dueling optimization methods on synthetic functions and real datasets, as well as conduct t-tests between the best-performing method and the runner-up with a significance level of 0.05.

4. We have redrawn all the diagrams and experimental figures in the paper to make them more readable. At the same time, we have provided more detailed explanations for some of the figures, especially Figure 4.

5. We have added a notation section for the methodology section in the Appendix E.





**We want to systematically explain our experimental design:**

First, we conduct experiments on a series of synthetic functions and real-world tasks to verify the effectiveness and superiority of RAPBO.

1. **Experiments on synthetic functions.** To evaluate the performance of RAPBO, we conduct experiments in various types of optimization problems (see Figure 2).

2. **Experiments on real-world tasks.** To further explore the performance of RAPBO and its applicability to real-world tasks, we also test RAPBO on real-world tasks such as motion control and spacecraft trajectory optimization (see Figure 3).

Furthermore, to make our experimental section more comprehensive and to better showcase the performance of RAPBO, we have made additional efforts.

1. **Deeper exploration of the preference propagation technique.** To gain a deeper understanding of the preference propagation technique we proposed, we conduct an in-depth exploration of the preference propagation process (see Figure 4). The experimental results show that RAPBO not only captures additional preferences from the existing dataset but also ensures that the newly added preferences maintain a high accuracy rate.

2. **Fill the optimization performance gap.** To verify that our method indeed fills the optimization performance gap between preference based and function value based methods, we compare both types of methods on real-world datasets (see Figure 5). The experimental results show that under the same cost budget, RAPBO is competitive with, or even surpasses, function value based Bayesian optimization methods in terms of optimization performance.

3. **Hyper-parameter analysis.** We also conduct hyper-parameter experiments (see Figures 6, 7) to showcase the impact of hyper-parameters.

---

### Note · Authors · 2025-01-24

I have read and agree with the venue's withdrawal policy on behalf of myself and my co-authors.